# Optimizing cloud motion estimation on the edge with phase correlation and optical flow

Bhupendra A. Raut[1,2], Paytsar Muradyan[2], Rajesh Sankaran[1,2], Robert C. Jackson[1,2], Seongha Park[1,2], Sean Shahkarami[2,3], Dario Dematties[1,2], Yongho Kim[1,2], Joseph Swantek[1,2], Neal Conrad[1,2], Wolfgang Gerlach[1,2], Sergey Shemyakin[2,3], Pete Beckman[1,2], Nicola J. Ferrier[1,2], and Scott M. Collis[1,2]

[1]Northwestern-Argonne Institute of Science and Engineering, Northwestern University, Evanston, Illinois, USA.
[2]Argonne National Laboratory, Lemont, Illinois, USA.
[3]University of Chicago, Chicago, Illinois, USA

**Correspondence:** Bhupendra Raut (braut@anl.gov)

**Abstract.** Phase Correlation (PC) is a well-known method for estimating cloud motion vectors (CMV) from infrared and visible spectrum images. Commonly phase shift is computed in the small blocks of the images using the fast Fourier transform. In this study, we investigate the performance and the stability of the block-wise PC method by changing the block size, the frame interval, and combinations of red, green, and blue (RGB) channels from the total sky imager (TSI) at the United States Atmospheric Radiation Measurement user facility's Southern Great Plains site. We find that shorter frame intervals, followed by larger block sizes, are responsible for stable estimates of the CMV as suggested by the higher autocorrelations. The choice of RGB channels has a limited effect on the quality of CMV, and the red and the grayscale images are marginally more reliable than the other combinations during rapidly evolving low-level clouds. The stability of CMV was tested at different image resolutions with an implementation of the optimized algorithm on the Sage cyberinfrastructure testbed. We find that doubling the frame rate outperforms quadrupling the image resolution in achieving CMV stability. The correlations of CMV with the wind data are significant in the range of 0.38–0.59 with a 95% confidence interval, despite the uncertainties and limitations of both datasets. A comparison of the PC method with constructed data and the optical flow method suggests that the post-processing of the vector field has a significant effect on the quality of the CMV. The raindrop-contaminated images can be identified by the rotation of the TSI mirror in the motion field. The results of this study are critical to optimizing algorithms for edge-computing sensor systems.

## 1 Introduction

Converting cloud images captured by a ground-based sky-facing camera into a time series of motion vectors has implications for reporting local weather and short-term forecasting of solar irradiance (Jiang et al., 2020; Radovan et al., 2021). Phase Correlation (PC) estimates global translative shift between two similar images by detecting a peak in their cross-correlation matrix which is used to estimate the cloud motion vectors (CMV) from the satellite and ground-based sky camera images (Leese et al., 1971; Dissawa et al., 2017, 2021; Zhen et al., 2019; Huang et al., 2011). On the other hand, optical flow (OF) estimates the pixel-wise motion assuming the conservation of brightness of the object pixels in two frames (Apke et al., 2022; Mondragón

et al., 2020; Peng et al., 2016). However, OF is sensitive to image noise and the variation in lighting. Both OF and PC methods fail to detect texture-less motion. Other object-based cloud tracking methods used in radar and satellite meteorology require

cloud identification before the tracking stage. The cloud identification approaches vary from threshold-based to texture-based methods and machine learning methods (Steiner et al., 1995; Raut et al., 2008; Park et al., 2021).

The texture-based methods and the machine learning models add computational overhead complicating their use in real-time applications. In infrared and microwave satellite images, and radar images, the threshold of brightness temperatures and reflectivity, mark a physical distinction of the features in the scene. However, for the cloud images in the visible spectrum,

thresholds of RGB values may not be a meaningful criterion to distinguish the properties of the clouds because they are affected by the lighting conditions and time of the day. The texture-based techniques are also susceptible to detection errors due to reflections and shadows caused by solar zenith angles. While the optical flow (OF) method estimates dense motion field (Horn and Schunck, 1981; Chow et al., 2015), it also suffers from the limitations in visible camera images and may require segmentation or background subtraction before the images are processed (Denman et al., 2009; Wood-Bradley et al., 2012;

El Jaouhari et al., 2015).

The Sage Project is designing and building a new kind of reusable cyberinfrastructure composed of geographically distributed sensor systems (Sage Waggle nodes shown in Figure 1a) that include cameras, microphones, and weather and air quality sensors generating large volumes of data that are efficiently analyzed by an embedded computer connected directly to the sensor at the network edge (Beckman et al., 2016, https://sagecontinuum.org/). An edge device rapidly analyzes the data in

real-time at the location where it is collected, and continuously sends and receives feedback from connected remote computing systems and other similar devices. In such networks including Sage, the computational efficiency of the algorithm is critical. The PC method can be implemented without preprocessing images and is robust to noise and changes in illumination as it works by only correlating the phase information (Chalasinska-Macukow et al., 1993; Turon et al., 1997). This eliminates the burden of separating the background from the objects to be tracked. A straightforward implementation of the PC method in the

frequency domain using the fast Fourier transform (FFT) is computationally efficient, and hence a natural choice to detect the cloud motion vectors from the hemispheric camera images at the edge.

The PC method is efficient for uniform rigid body motion, i.e. when an object's shape and size are preserved, and multiple objects in the scene are moving with the same velocity. There are a few limitations to the PC method that affect its applicability in tracking cloud motions in a sky-facing camera. First, the PC method is less efficient when multiple peaks in the correlation

matrix are observed. This occurs when cloud features are moving with different velocities as each peak is associated with the motion of one or more independent features in the images. This limitation is overcome by dividing the image into sufficiently smaller subregions or blocks and employing the PC separately for each block (Leese et al., 1971). As the multi-layer clouds with different cloud base heights move independently, Peng et al. (2016) used adaptive blocks for each cloud type.

Second, the changing cloud texture and geometries may cause incoherent motion vectors in some image blocks. Therefore,

additional quality control measures are applied to remove the spurious CMVs, usually assuming that a spurious CMV substantially deviates from its surrounding CMVs in the presumably smooth velocity field (Westerweel and Scarano, 2005). For

the assumption of the coherent velocity field, smaller block sizes are preferred. The optimal block size is determined by the maximum expected displacement during the frame interval.

Third, the ground-based cameras frequently encounter contamination on the mirror dome or hemispherical lens, obscuring the clouds during and after a precipitation event and automated identification and removal of precipitation-contaminated images are critical (Heinle et al., 2010; Kazantzidis et al., 2012; Gacal et al., 2018; Voronych et al., 2019). The distortion of images caused by the presence of raindrops and the edge detection methods are used to identify raindrop contamination (Kazantzidis et al., 2012; Voronych et al., 2019). In this paper, we propose the use of motion vectors for detecting raindrop contamination on the rotating TSI mirror.

Finally, while it is common for cameras to produce high-resolution three-channel images, the PC method utilized only a single channel. Hence, either the grayscale image or one of the RGB channels is used. The dependence of CMV stability on the choice of image channels is undocumented.

Investigating the sensitivity of the motion vectors to the block sizes, the frame frequency, and its response to different spectral channels will help in the effective implementation of the method. Therefore, in this paper, we evaluate the performance of the block-wise PC with three visible channels, the grayscale, and the red to the blue ratio in two block sizes and two frame rates. We also demonstrate the effect of change in the image resolution and the change in frame rate on the CMV quality. We also validated the PC method with constructed data and compared it with OF method. The wind and ceilometer measurements are used for additional validation to show consistency with independent atmospheric measurements. However, wind retrieval is not an objective of the paper. The data, methodology, and algorithm are described in section 2. The results are shown in section 3, and their implications for the Sage edge-computing platform are discussed in section 4.

## 2 Data and Methods

### 2.1 Data

In this paper, we mainly used data from the Atmospheric Radiation Measurement (ARM) user facility's Southern Great Plains (SGP) atmospheric observatory (36.7°N, 97.5°W), in particular, at the supplemental S1 and central C1 facilities in Lamont, OK, due to long-term data availability from colocated instruments for wind and cloud base height measurements. The Sage camera images are used in section 3.3.2.

#### 2.1.1 Total Sky Imager

The Total Sky Imager (TSI) is a mounted full-color digital camera looking downward toward a rotating hemispherical mirror (Figure 1b). Daytime full-color hemispheric sky images are obtained from TSIs operational at the ARM SGP atmospheric observatory (Morris, 2005; Slater et al., 2001). The images recorded over the S1 site every 30 seconds (Morris, 2000) during the day on July 26, 2016, are used to demonstrate the sensitivity of the method described later on. The central sky region of

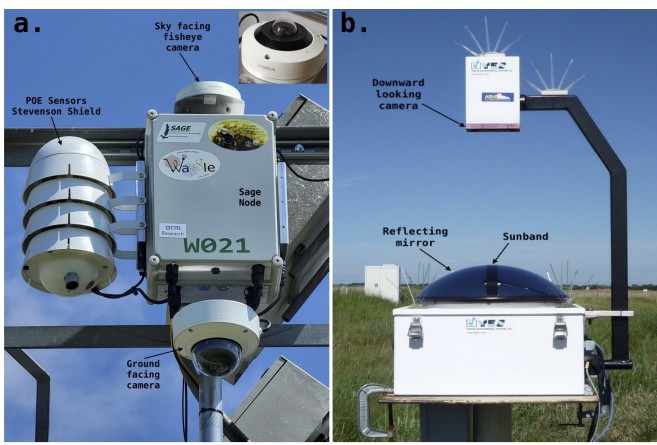

**Figure 1.** a) Sage node deployment at the ARM user facility in Lamont, OK., with a fisheye camera for sky monitoring. b) Downward-looking Total Sky Imager with rotating mirror sunband and setup.

$400 \times 400$ pixels is used to compute the CMV during the 06:36 to 20:35 CDT window. The data over the C1 site between October 14, 2017, and August 14, 2019, are used for comparison of CMVs with the wind data.

### 2.1.2 Sage Camera

Hanwha Techwin America's fish-eye camera (XNF-8010RV X series), hosted atop a Sage node and pointed toward the sky at the Argonne Testbed for Multiscale Observational Studies (ATMOS) (41.70°N, 87.99°W), has a 6 MP CMOS sensor providing $2048 \times 2048$ pixels full-color images. Unlike the TSI camera, the Sage fish-eye camera lacks a sunband and a rotating mirror (Figure 1). Images recorded from this camera every 30 seconds from 06:00 to 17:00 CDT on February 13 and 14, 2022, are used to demonstrate the effect of camera resolution and frame rate on the sensitivity of the method.

### 2.1.3 Wind Profiling Radar and Ceilometer

To validate the estimates of the CMV in our work, cloud base height (CBH) and wind measurements are obtained from the co-located ceilometer and the wind profiling radar (WPR), respectively (Muradyan and Coulter, 1998; Morris et al., 1996). The ceilometer is an autonomous, ground-based active remote sensing instrument, that transmits near-infrared pulses of light and detects multi-layer clouds from the signal backscattered from cloud droplets that reflect a portion of the energy back toward the ground. (Morris, 2016). The laser ceilometer measurements extend up to 7.7 km with 10 m vertical resolution. The wind profiles for comparison were obtained from the 915 MHz WPR, which transmits electromagnetic pulses in vertical and multiple tilted directions (3-beam configuration is used at SGP) to measure the Doppler shift of the returned signal due to atmospheric turbulence from all heights (Muradyan and Coulter, 2020). The consensus-averaged winds are estimated at an hourly interval

and are available from 0.36 km to about 4 km at 60 m vertical resolution. We used the CBH and wind estimates over the SGP C1 site from October 14, 2017, to August 14, 2019.

## 2.2 Phase Correlation using FFT

The phase correlation method for estimating motion in the images is based on a property of the Fourier transform that a translational shift in two images produces a linear phase difference in the frequency domain of the Fourier transform of the images (Leese et al., 1971). In other words, a signal $f_2$ that is related to signal $f_1$ by a translation vector $(d_x, d_y)$, then their Fourier transforms denoted by $F_1$ and $F_2$ have equal magnitudes but with a phase shift of related to the normalized cross power spectrum as follows.

$$e^{-i2\pi(\mu d_x + \nu d_y)} = \frac{F_1(\mu,\nu)F_2^*(\mu,\nu)}{|F_1(\mu,\nu)F_2(\mu,\nu)|} \tag{1}$$

where $F_2^*$ is the complex conjugate of $F_2$. The phase shift term $e^{-i2\pi(\mu d_x + \nu d_y)}$ is the Fourier transform of the shifted Dirac delta function. Hence, we can calculate $d_x$ and $d_y$ by computing the inverse Fourier transform of the cross-power spectrum and finding the location of the peak (Leese et al., 1971; Tong et al., 2019). Therefore, PC in small image blocks, between the subsequent images, is rapidly computed using FFT. Because the phase correlation is executed only for a small image block, it is possible to employ parallel computation to further speed up the estimation of motion for a large dataset.

The following procedure describes the steps in implementing PC to estimate the shift in images $I_1(x,y)$ at time $t_1$ and $I_2(x,y)$ at time $t_2$. Let image $I_2$ be spatially translated by $d = (d_x, d_y)$ with respect to the image $I_1$,

1. Obtain FFT of the images $I_1(x,y)$ and $I_2(x,y)$ as $I_1(\mu,\nu)$ and $I_2(\mu,\nu)$.

2. Compute $C(\mu,\nu)$ by multiplying the FFT of the first image and the complex conjugate of the second image. $C(\mu,\nu)$ is the cross-covariance matrix in Fourier space.

3. Obtain an inverse FFT of $C(\mu,\nu)/|C(\mu,\nu)|$. The real part of the outcome gives a covariance matrix $Cov(p,q)$ in image space.

The above implementation of the PC algorithm is available in several programming languages, notably C++, Python, and R in packages openCV (mulSpectrums), SkImage (phase_cross_correlation), and imagefx (pcorr3d). For this study, we used a custom Python implementation same as Picel et al. (2018); Raut et al. (2021). (See code availability section).

If image $I_2$ is a spatially translated version of the image $I_1$, then the phase covariance matrix $Cov(p,q)$ is zero everywhere except for a sharp peak at the location corresponding to the displacement between the two images. The peak intensity is a good measure of the quality of the motion vector. Due to the reasons mentioned in Section 1, the actual peak in the covariance matrix can be fuzzy and it corresponds to the best-fitting translational motion in the images. Sharp single-pixel peaks can sometimes occur in the covariance matrix, due to the high-frequency noise and artifacts in the images, which are flattened using Gaussian smoothing on Cov(p,q) with $\sigma = 3$. An example of the procedure is given in Raut et al. (2021).

For each image block, the peak covariance location is assigned as the local motion vector in image $I_2$ with reference image $I_1$. As per the meteorological convention for winds, the U component is positive for an eastward flow, and the V component is positive for a northward flow. The location of the peak covariance from the center of the matrix gives the shift in the image features during the image interval along the X, and Y dimensions of the image. We saved X and Y shifts and computed the motion vectors per minute. The image top is oriented towards the north and therefore in the subsequent sections, the motion in the X and Y directions are referred to as U and V components, respectively.

## 2.3 Constructed Data for Validation

For studying the accuracy and quantitative error analysis of the method, a dataset with the known displacement vectors is needed. Synthetic or reconstructed image sequences are best suited for this task as managing the displacement is trivial in such a dataset compared to the real dataset. However, the constructed dataset should be made with care to avoid unreal augmentations and artifacts while incorporating possible variations of the features from image to image. Such a dataset, although possibly not a perfect representation of the real data, can be used to study the properties of the algorithms.

These images can then be translated by the desired amount to achieve the cloud motion effect. We created image pairs by reconstructing the 2060 samples of Sage camera images classified as cloudy by the algorithm described in Dematties et al. (2022) in their cluster 3 and 8. The images were selected to have cloudiness in the central $200 \times 200$-pixel region. The pair of images were created and then subjected to the following modifications using an edge filter $A$ and a flat filter $B$.

$$\text{Kernel } A = \begin{bmatrix} 0 & -1 & 0 \\ -1 & 5 & -1 \\ 0 & -1 & 0 \end{bmatrix} \tag{2}$$

$$\text{Kernel } B = \begin{bmatrix} 1 & 1 & 1 \\ 1 & 1 & 1 \\ 1 & 1 & 1 \end{bmatrix} \tag{3}$$

The first image was created with the following operations.

1. The original image was converted to grayscale.

2. Addition of Gaussian noise with mean zero and standard deviation 1.

3. Convolution with Kernel A.

4. Two iterations of Erosion followed by Dilation by the Kernel B. i.e., Morphological opening of the image.

5. Cropped images to achieve the desired displacement.

The second image of the pair was created by modifying a few operations.

1. Reversed the RGB colors in the original image before converting it to grayscale. This reversing of operations also known as color augmentation creates a spectrally different image with the same structure.

2. Addition of Gaussian noise with mean zero and standard deviation 1.

3. Convolution with Kernel A.

4. One iteration of morphological opening by Kernel B.

5. Translated and cropped images for the desired displacement.

We translated the images by 5, 10, and 20 pixels in both X and Y directions for ease of comparison and interpretation of the results (see section 3.1).

## 2.4 Outliers in the CMV Field

When the image block belongs to the clear sky or the scene has changed beyond recognition by the correlation, the peak in the covariance matrix is usually near the boundaries of the block, thus giving artificially large displacements. Such vectors are easily identified using a maximum velocity limit $V_{max}$. For this analysis, we used $V_{max} = \frac{block\ length}{3}$. If the $V_{max}$ is smaller than the expected maximum speed, then a larger block size is recommended.

Removing large magnitude vectors smooths the field, however some motion vectors of reasonable magnitude but spurious directions remain. Such spurious vectors can be removed by comparing them with the surrounding motion vectors.

We compared each vector with the normalized median fluctuation of the neighboring blocks (Westerweel and Scarano, 2005). Consider a 3×3 data with $u_0$ as the displacement vector at the center block, $u_1, u_2, ..., u_8$, as displacement vectors of the neighbors, and $u_m$ as the median of neighbors, not including the central vector. Then the residual ($r_i$) of all neighbours are computed as $r_i = |u_i - u_m|$ to obtain the median residual ($r_m$). The normalized median fluctuation $r_0$ is given by

$$r_0 = \frac{|u_0 - u_m|}{r_m + \epsilon} \tag{4}$$

$\epsilon$ is the minimum normalization level that represents the acceptable fluctuation, usually 0.1–0.2. The CMV vectors with normalized median fluctuation values over 6 are discarded as outliers.

## 2.5 Identification of Raindrop Contamination

The CMV is not valid when rainwater present on the reflecting mirror obscures the clouds. However, in such a scenario, the rotation of the raindrop-contaminated mirror produces a rotating vector field as shown in Fig. 2a. We correlated the estimated CMV fields with the mean of manually identified contaminated CMV fields and found that the correlation coefficient, $r > 0.4$ is associated with the rotation of the raindrop-contaminated mirror (Fig. 2b). Because of the sharp edges of the raindrops, the

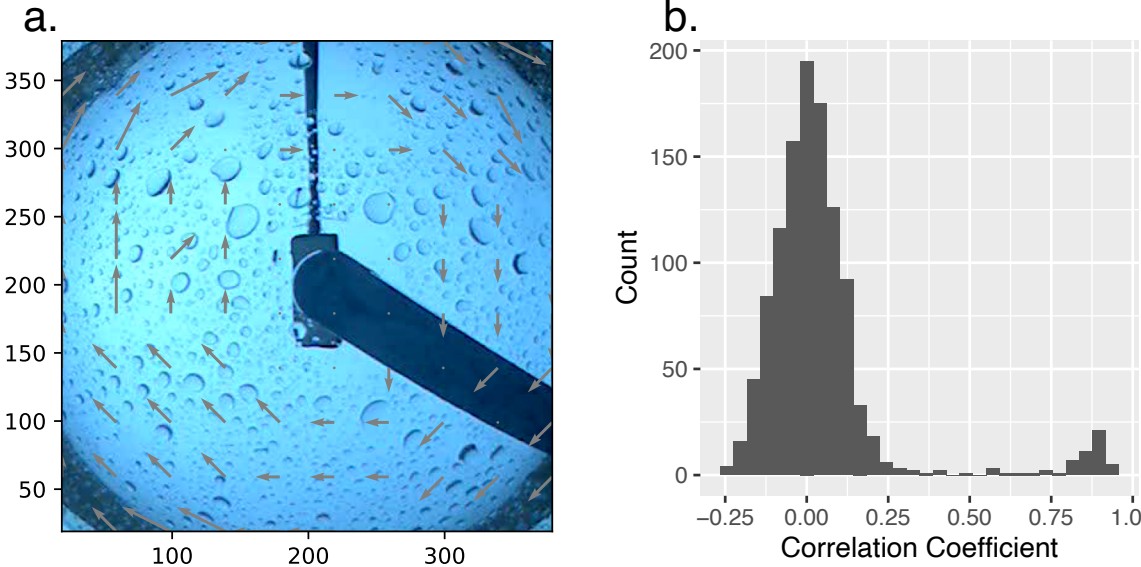

**Figure 2.** a) An example of the circular motion field generated every 2-4 minutes by the rotation of the raindrop-contaminated mirror of TSI. b) Histogram of the correlation coefficient between the mean rotational vector field and CMV fields on January 2, 2017, shows a robust separation of raindrop-contaminated frames from the clean frames.

rotational pattern is efficiently captured with few raindrops contaminating the mirror. However, it struggles to detect contamination when the drops are concentrated at the center of the dome. Therefore, after the rotation is detected, the next 10 minutes of data are flagged as contaminated even if no subsequent rotation is detected.

## 2.6 Setup for Sensitivity Analysis

To test the algorithm's sensitivity to the block size, we divided the $400 \times 400$-pixel sky area into a grid of $10 \times 10$ and $20 \times 20$ blocks and referred to as block length 40 and 20 pixels, respectively in Figures 5–8. Note that the choices for the number or size of blocks are restricted by the $V_{max}$ on one end and the neighborhood criteria on the other. For example, if the expected $V_{max}$ is 7 pixels/min then the blocks should be at least 21 pixels wide (section 2.4). On the other hand, for the $10 \times 10$ grid (block width 40 pixels) with a one-pixel neighborhood, the correction applies to the central region of $8 \times 8$ blocks only. Therefore,

increasing block sizes reduces the number of blocks in the sky region, which reduces the scope of the neighborhood method in the error correction stage. To test the sensitivity to the frame interval, CMVs are also computed at 30 and 60-second intervals. The 30-second CMVs are accumulated over one minute for comparison. As the PC uses monochromatic images, the CMVs were computed separately for the three BGR channels (abbreviated to Bu, Gn, Rd in Figures), the red to the blue ratio (RB, Slater et al., 2001), and grayscale (Gy) images.

**Table 1.** Mean, standard deviation (SD), root mean square error (RMSE), and root mean square percent error (RMSPE) of cloud motion estimated from reconstructed images for constant displacements of 5, 10, and 20 pixels. (u=uncorrected, c=corrected with a threshold.)

| Displacement [pix] | Mean | SD | RMSE | RMSPE % |
|---|---|---|---|---|
| 5 (u) | 5.5 | 1.02 | 1.13 | 22.6 |
| 10 (u) | 8.2 | 2.63 | 3.19 | 31.9 |
| 20 (u) | 15.4 | 8.7 | 9.83 | 49.1 |
| 10 (c) | 9.2 | 1.4 | 1.7 | 11.4 |
| 20 (c) | 20.5 | 2.1 | 2.1 | 4.4 |

## 2.7 Optical Flow Algorithm for Comparison

Let $I(x, y, t)$ be the first image defining the pixel intensities at the time $t$. Therefore, the first and second images are related as

$$I(x, y, t) = I(x + \delta x, y + \delta y, t + \delta t) \tag{5}$$

In the computation of OF, we assume that the intensities of the pixels, that belong to the exact object, change only due to the displacement (Horn and Schunck, 1981). This assumption allows for all changes detected in the x and y directions of the image are to be associated with the motion only. The first-order approximation of the Taylor polynomial is

$$\frac{\partial I}{\partial x} u + \frac{\partial I}{\partial y} v + \frac{\partial I}{\partial t} = 0 \tag{6}$$

where $u = \frac{dx}{dt}$, $v = \frac{dy}{dt}$. However, to find the dense motion vector field, we used Farnebäck (2003) method from OpenCV which approximates the neighborhood of both frames by higher order (quadratic) polynomials, $I(x) \sim x^T A x + b^T x + c$. This algorithm works with an image pyramid with a lower resolution at each next level to track the features at multiple resolutions. Faster motions are captured with the increased levels of the pyramid. The algorithm provides a motion vector for each pixel of the input image. The motion field can be smooth or detailed depending on the given neighborhood size and the standard deviation used for the polynomial expansion.

## 3 Results

### 3.1 Validation with Constructed Images

To show the validation of our implementation of the PC method, we used the images reconstructed from the Sage camera data as described in section 2.3. Finally, 2060 pairs of cloudy images translated by 5, 10, and 20 pixels, in both X and Y directions, were used to estimate the displacement using the PC method described in the section 2.2. The distributions of the estimated motion are shown in Figure 3 and their comparison statistics are shown in Table 1. For smaller displacement of 5 pixels, the algorithm estimates the values with 22.6% root mean square percent error. With the increasing displacement of 10

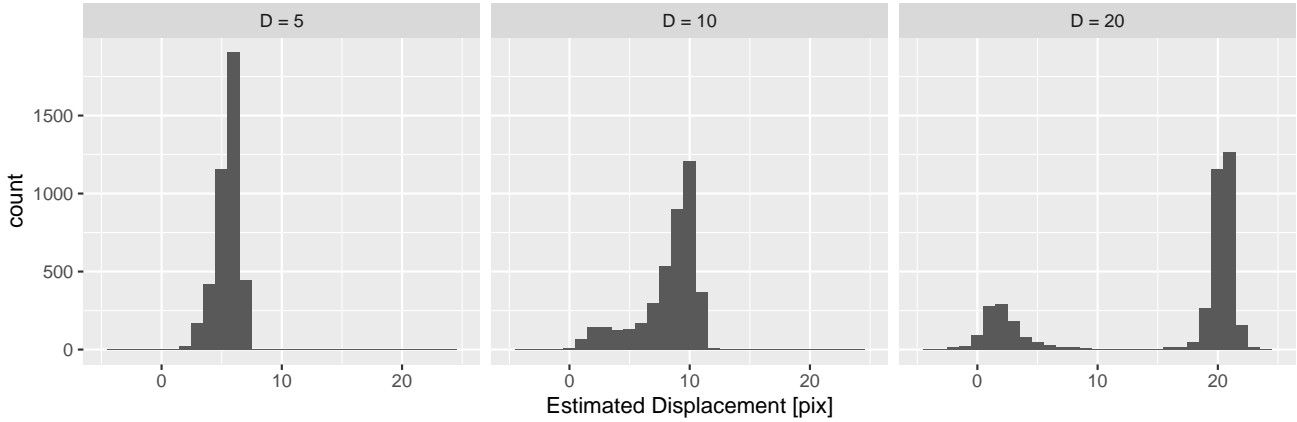

**Figure 3.** Distribution of the motion estimated by the PC method in reconstructed images for displacement values 5, 10, and 20 pixels.

and 20 pixels, the RMSPE increases to approximately 32 and 49 %, respectively. This is consistent with the increasing spread in the estimates with increasing displacement as seen in Figure 3. However, the algorithm tends to produce a peak near the zero value, except for very small displacements (D=5), and another peak at the given displacement. These results are consistent with Zhen et al. (2019). The proportion of vectors near zero value increase with the displacement however, in most cases they are estimating the correct quadrant of the direction of the motion. However, these values need to be removed to get a good estimation of the speed of the motion. For demonstration purposes, we used the threshold to remove the near-zero values which significantly reduced the RMSE. However, in the real images, the method described in the section 2.4 is effective when the majority of the vectors are correct. For D=20, approximately a quarter of the vectors were near zero vectors.

## 3.2 Cloud Motion and Sensitivity Results

Changing sky conditions captured by TSI on July 26, 2016, during the 06:36 to 20:35 CDT are shown in Fig. 4 at 100 minutes intervals for reference. The sequence of images shows the movement of stratiform clouds from the southwest for over two hours ($\sim$ 150 min), with the occasional presence of low-level cumulus clouds. After about 3 hours, the cumulus cloud development covered the sky (see the 200-minute snapshot) moving predominantly from the east/northeast, as shown by the red arrow. Rapidly moving low-level clouds had less coherent motion at the block level than the altostratus. In addition, the low-level clouds intermittently traveled in patches with the altostratus aloft moving from the southwest. The time series of U and V components of CMV, shown in Fig. 5 and Fig. 6, respectively, are smoothed using cubic splines for easily discernible visualizations. The raw U component is shown in Fig. 12 for reference. The U and V plots suggest that the PC method successfully captured the direction of the motion and the reversal of the direction in all configurations. As described above, the mid-level clouds moving from the west and transition to low-level clouds moving from the east at around 150 minutes are seen in Fig. 5.

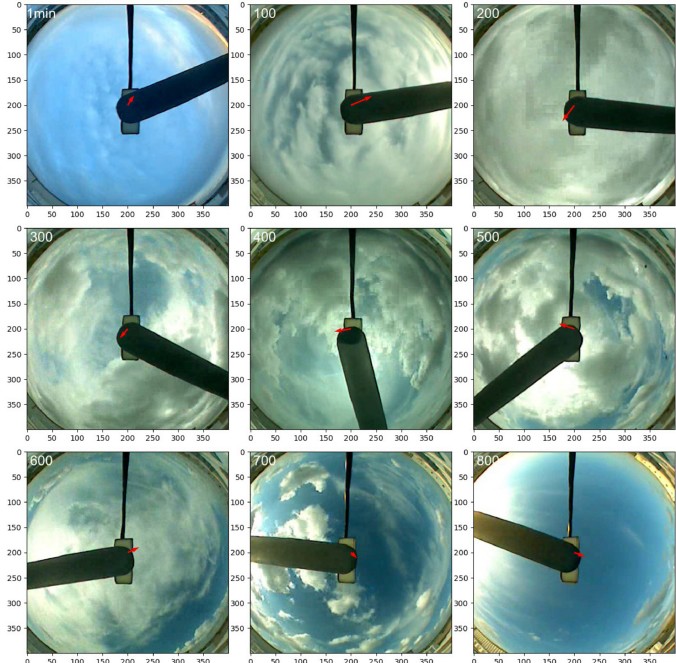

**Figure 4.** Varying sky conditions on 26 July 2016, from 06:36 to 20:35 CDT (11:36-01:35 the next day in UTC) at 100 minutes intervals over Lamont, OK. A Sky area of 400x400 pixels is cropped and used for CMV computation. The top of the image point to the north and the red arrow shows the direction of motion for that frame.

The turbulent motion characterized the episodes of cumulus growth from 150 to 450 minutes, as evidenced by the fluctuations in the CMV during this phase in all channels, however, more pronounced in the RB channel. Between 500 and 600 minutes, cumulus and altostratus cleared, and high-level cirrus clouds became visible, flowing from the west. Additional late-afternoon cumulus movement (see the 700 min snapshot) and the clear sky with high-level cirrus or occasional westward-moving low-level cloud patches were present until sunset.

The frequency distribution of the CMV components (Fig. 7) also shows two peaks of positive eastward component (U) distinguishing the rapidly moving mid-level and slow high-level clouds from the camera viewpoint. The larger blocks (40 pixels wide) and the shorter frame interval (30-sec) have a wider range than the rest of the configuration, which shows their efficiency at capturing the low-level cumulus motion. It is important to note that July 26, 2016, was accompanied by a variety of cloud conditions and individual episodes of low, medium, and high-level cloud motion, each lasting for at least an hour. Thus, the short-term fluctuations of CMV are mainly caused by the algorithm's instability. To assess the stability of CMVs for various configurations, we compare the autocorrelation of the CMV in the following subsection.

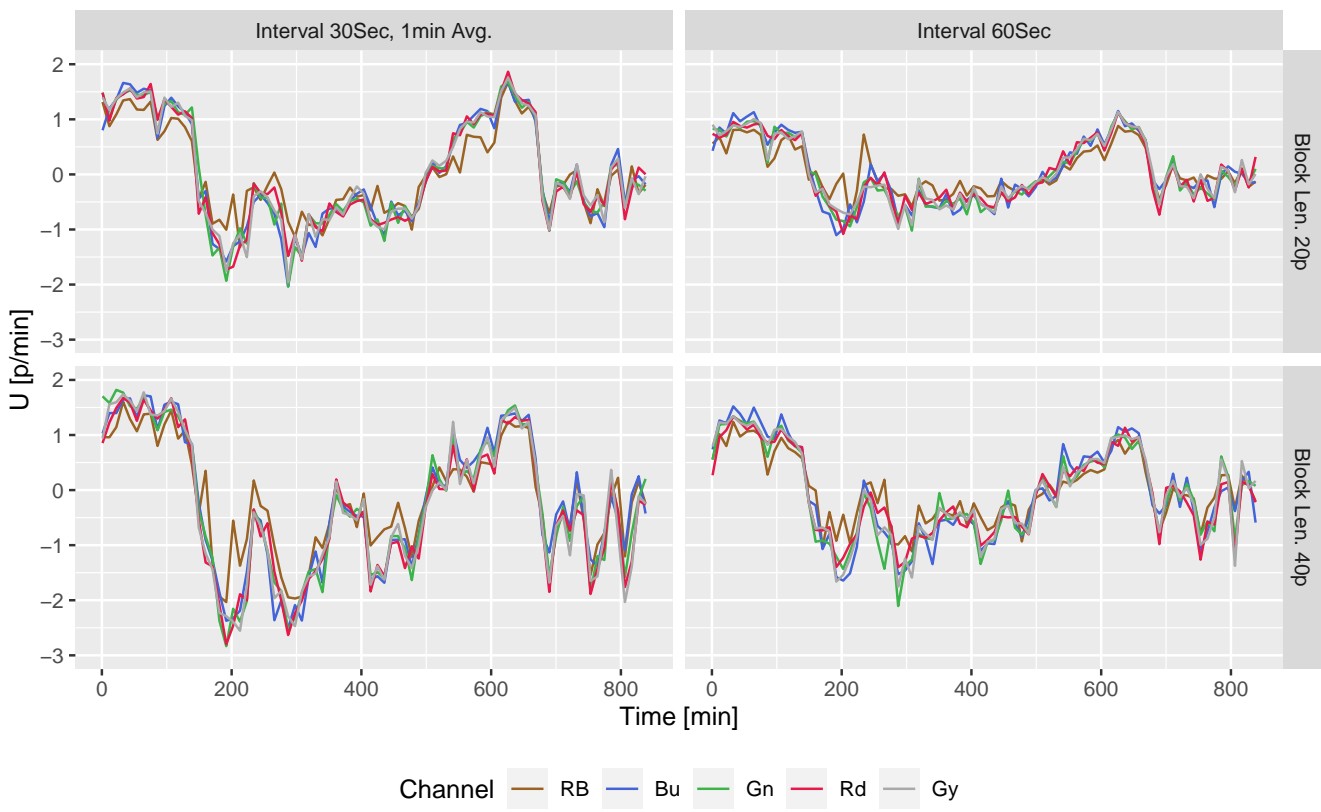

**Figure 5.** Smoothed time series of U component of domain averaged CMV [$\mathrm{pixel \cdot min^{-1}}$] on 26 July 2016, 06:36 to 20:35 CDT (11:36-01:35 next day in UTC) over Lamont, OK. Variations with block size (20 pixels and 40 pixels) and frame intervals (30 sec and 60 sec) are shown for 5 channels.

## 3.3   Stability of CMV

The stability of the CMV was tested by changing the block size, the frame interval, and combinations of red, green, and blue (RGB) channels from the total sky imager (TSI) and by changing the image resolution and frame rate in the Sage camera.

### 3.3.1   Block Size, Frame Interval and Channel

The movement of clouds is usually smooth at the one-minute time interval. Except for the change in direction during the altostratus to cumulus transition, the movement of the clouds on July 26, 2016, should be more or less stable at the hourly intervals for most of the day (Fig. 5 and 6). However, the CMV fluctuates at a 1-minute time interval, mainly due to the irregular response of the algorithm caused by the issues mentioned in Section 1. Therefore, the stability of motion vectors in time is evaluated for the above configurations by checking the autocorrelation of the CMV time series. The autocorrelation function (ACF) of U and V components for different configurations is shown in Fig. 8 (top panels). The linear ACF suggests a

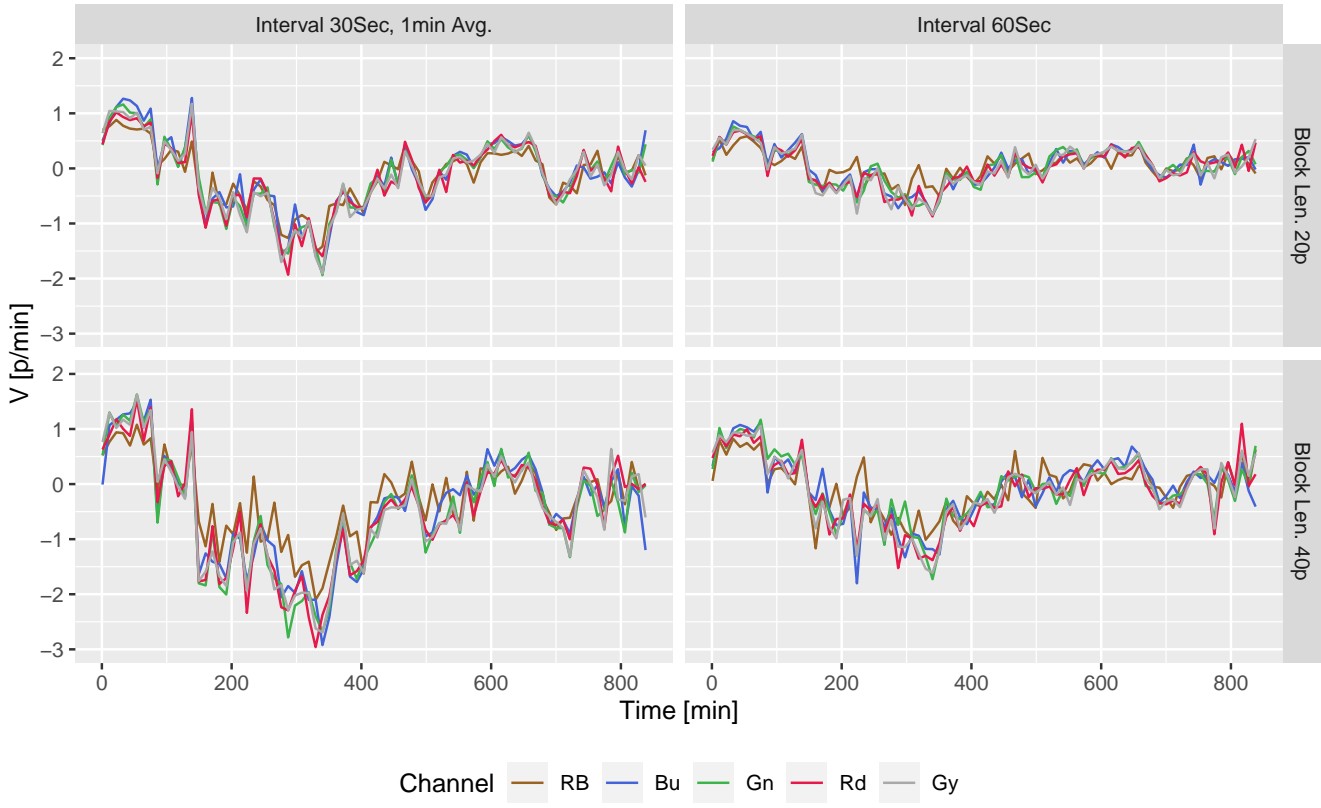

**Figure 6.** Same as Figure 5 but for V component of the CMV.

long decorrelation length for all the combinations. While RB has the lowest autocorrelations (more fluctuating vectors) for all configurations, the rest of the color channels have more or less equally stable vectors. The frame interval, followed by block length, noticeably affects the stability of the vectors.

The lower panels in Fig. 8 are the same as the top panels but for the period between 150:450 minutes when the rapidly developing low-level clouds were present. The small cloud features were developing fast and had variable motion. Therefore, during this period, the autocorrelation is lower and the performance of the large block sizes and short frame intervals is noticeably better for both U and V components. The CMV from red and gray channels has slightly higher autocorrelation for the dominant motion (i.e. zonal component, U) during this period.

**3.3.2    Image Resolution and Frame Interval**

Our analysis shows that CMVs are more stable for larger blocks and shorter frame intervals (see Sec. 3.3.1). Therefore, the stability of motion vectors is evaluated for the same blocks (i.e., the image divided into $10 \times 10$ grid.) and by reducing their resolution in steps to block lengths of 200, 150, 100, and 50 pixels, as shown in Fig. 9, with frame intervals of 30 and 60

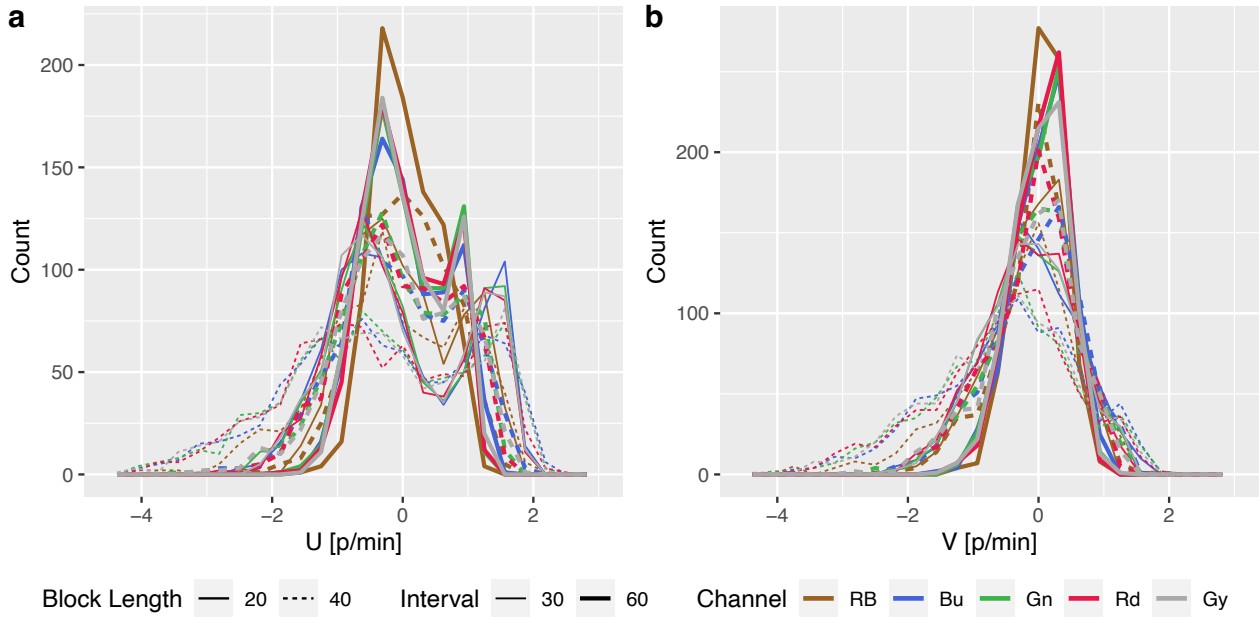

**Figure 7.** Frequency distributions of U and V components [pixel · min$^{-1}$] shown in Figures 5 and 6, respectively for all 20 setup combinations. The bimodal distribution of the U component is due to two cloud regimes discussed in sec

3.2

seconds. February 13 was dominated by mid-level stratus cloud motion and February 14 had periods of low-level cumuliform
development with fast movements and rapid evolution of cloud features dominating the scene. In addition, on both days, the
cloud motion was mostly in East-West (zonal) direction with the U component approximately four times larger than the V
component. Therefore, ACF of only U components for four image resolutions and two frame intervals are shown in Fig. 10.
ACF is significantly lower for longer frame intervals. For example, long intervals reduce the autocorrelation at lag-1 from 0.75
at 30-sec intervals to 0.5 at 60-sec intervals (Fig. 10 a ). This effect is even more prominent for the rapidly evolving cumuliform
clouds (Fig. 10 b ) where the autocorrelation at the lag-1 drops from 0.65 to 0.2. On the other hand, a change in the resolution
by a factor of four has minimal effect, and a change in lag-1 autocorrelation is within 0.05.

### 3.4   Comparison with Wind Data

To compare the hourly mean CMV with winds of appropriate altitudes, we identified the hours with a stable CBH for at least
20 minutes from the ceilometer measurements from October 14, 2017, to August 14, 2019. The hourly winds are averaged
for 1 km deep layers from the surface to 4 km altitude, and then the hourly-mean CMVs are compared with the mean wind
vectors in the vertical layer corresponding to the median CBH (Fig. 11). Note that the range of values for CMV and wind
have an order of magnitude difference due to the different units. From the 551 days of data during this period, 876 daytime
cloudy hours were identified, when simultaneous measurements from the WPR, the ceilometer, and CMV estimates were

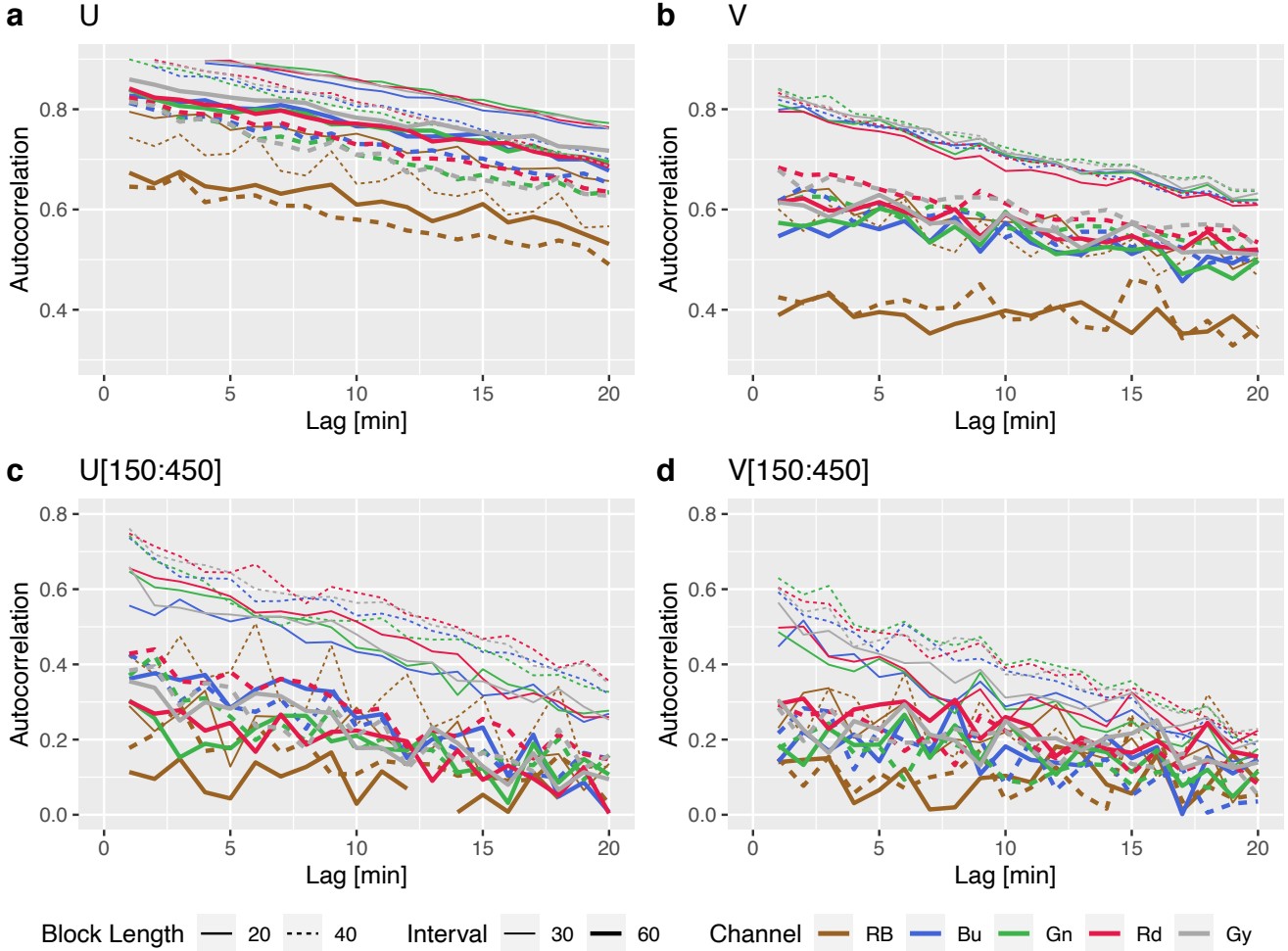

**Figure 8.** Autocorrelogram for U and V components showing the stability of the motion vectors shown in Figure 5 and Figure 6. (top) for all the data, and (bottom) for the selected period of rapid Cu cloud development between 09:06 to 14:56 LT (time steps: 150–500 in ).

available. We only present CMVs for one setting: the 40-pixel block length, and the 30-second frame interval for the red
channel. The rainy samples, identified using the method described in section 2.5 mostly fall close to zero value, as no mean
motion is recorded. The sky-view camera data routinely suffers from rain, snow, and other debris on the lens that obstructs the
view. The higher wind speeds near zero CMV can mainly occur due to the snow obstructing the view, or smooth flat cloud
bases that are not successfully tracked. In addition, the quality of the wind profiles from the WPR is also adversely affected
by rainfall (Muradyan and Coulter, 2020). Therefore, we removed instances with precipitating events from consideration in
our comparison. The correlation coefficient ($r$) of the U component of the CMV and hourly wind averages improved from
0.38 for all the data, to 0.42 after removing rainy samples, with a 95 % confidence interval. Likewise, for the V component,
$r$ increased from 0.56 for all data to 0.59, with a 95 % confidence interval. The slope of the linear fit for U components is

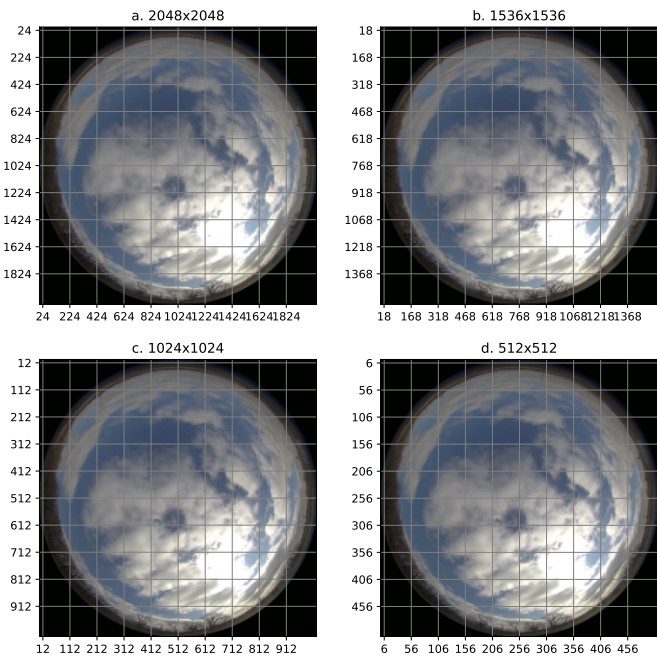

**Figure 9.** The scheme for testing resolution sensitivity with Sage camera image obtained on April 21, 14:06:38 over Lamont, OK. A $10\times10$ block grid with four successively lower resolutions is used for CMV computation to compare the effect of resolution and time interval on the stability of CMV.

between 2.4–3.4 for layers 0–3 km and it is 5.7 for the 3–4 km layer, suggesting that the mid-level (i.e. 3–4 km) CMVs are noticeably underestimated from the camera viewpoint. The slopes of the V components are in the range of 3–4 for all layers. The WPR data above 4 km are sparse hence no samples with matching criteria were available during the study period.

The comparison of the CMV either from a ground-based camera or satellite sensors with that of atmospheric winds has several sources of uncertainty. The estimation and comparison of CBH and winds from the ceilometer and the wind profiler respectively, show sampling uncertainty. In addition, the cloud displacement from the camera viewpoint differs with altitude, and deeper convective clouds do not always move parallel to the low-level winds. Therefore, this comparison may not be interpreted as a quantitative validation of the algorithm for wind retrievals, however, significant correlations of the magnitudes indicate that the estimates of the instantaneous CMVs from the camera images are stable over a long period. Although a perfect correlation does not exist between wind and CMV from ground camera images due to the above factors, more accurate identification of rain and snow-contaminated images would improve the comparison.

## 3.5 Comparison with Optical Flow Method

The estimations of the mean motion vector from PC and the OF algorithms for U components are shown in Figure 12. The issue of near-zero values seen in Figure 3 is also present in OF vectors which is causing an underestimation of the mean magnitude

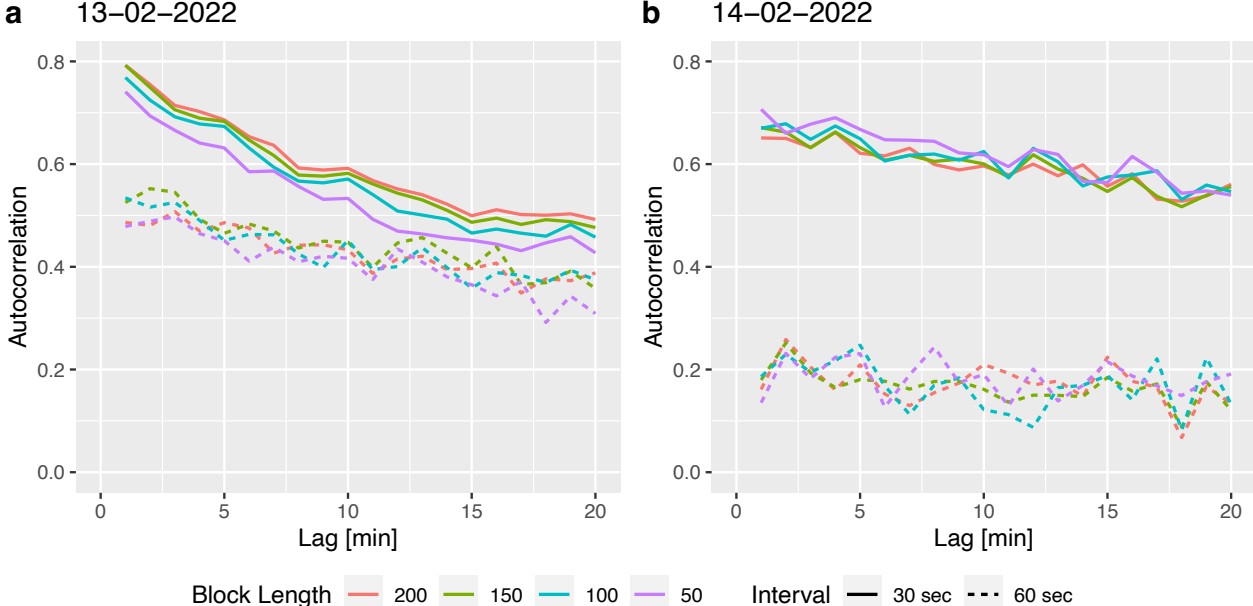

**Figure 10.** Autocorrelogram for U components for varying resolutions of the image with the same block region and the two frame intervals on February 13 and 14, 2022 shows the effect of the changing resolution and time intervals on the stability of the motion vectors.

as compared to the PC (Figure 12, OptFlowAll). Figure 13 shows smoothed dense CMV field using OF method. The near-zero values occur at the clear sky region or where the lighting and scene change drastically. Due to the dense motion field, these vectors are clustered in image space and therefore they can not be removed with the neighborhood method of Westerweel

and Scarano (2005). However, the regions with cloudiness are efficiently tracked by the OF method. After removing near-zero magnitudes using an arbitrary threshold of 1, the OF has higher magnitudes as compared to the PC method and better captures the variability than the PC method. The dense field of motion vectors can be leveraged for more adaptable statistical corrections than the arbitrary threshold used in this study for presentation purposes. The final CMV magnitudes could be highly dependent on the post-processing of the results for both PC and OF methods. Although the mean magnitudes are sensitive to post-

processing corrections, the change in direction and magnitude of the motion vectors from both methods are comparable. The correlation between the OF and PC methods increases from 0.84 to 0.9 after removing the near-zero values. The autocorrelation functions in Figure 12b show that the minute-by-minute fluctuations of the CMV are more stable for OF than for PC, due to the dense vector field of OF.

## 4    Discussion of the Results

Prior studies have documented the effectiveness of the block-wise PC and OF method for detecting cloud motion in IR and visible spectrum images (Leese et al., 1971; Chow et al., 2015; Dissawa et al., 2017; Zhen et al., 2019). We tested the sensi-

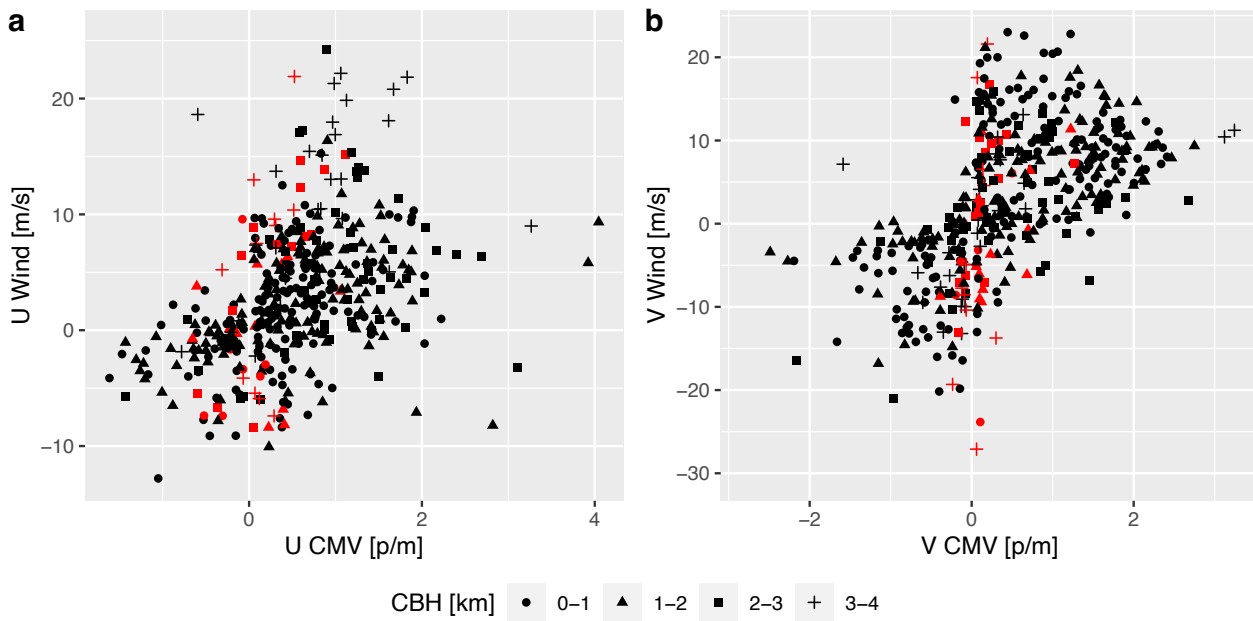

**Figure 11.** Comparison of hourly mean U and V components of the CMV and mean wind in a 1 km deep layer where the stable cloud base height was observed during the hour. The rainy hours extracted using the method in section 2.5 are shown with the red color.

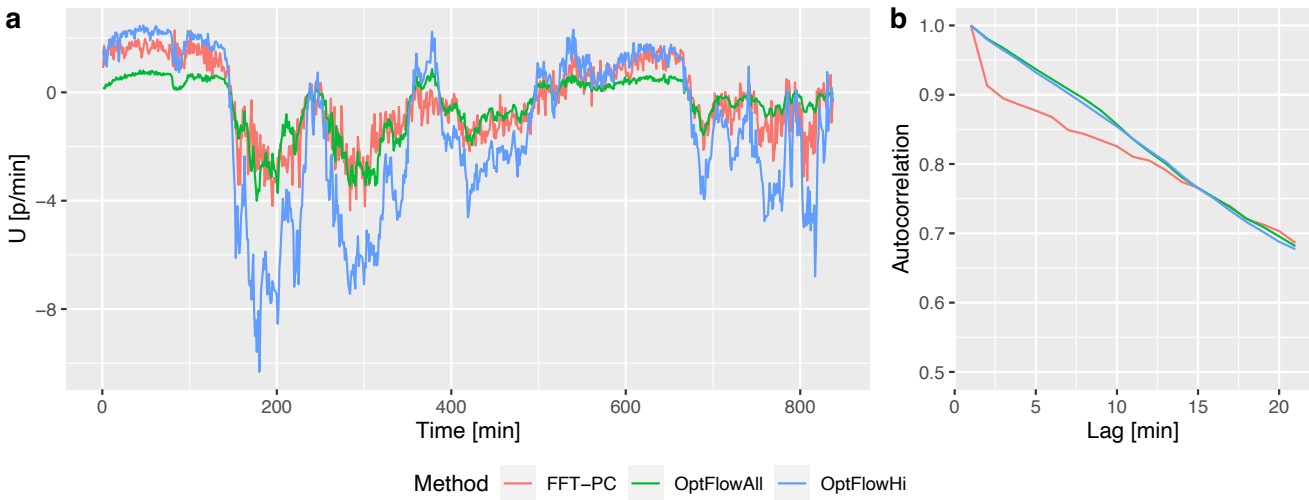

**Figure 12.** Comparison of mean cloud motion estimated using PC and OF methods on 26 July 2016, over Lamont, OK. The mean OF vectors computed with all valid data and after removing low values are shown with green (OptFlowAll) and blue (OptFlowHi) lines, respectively. a) time series of domain average U component taken at the central 400×400 pixel region at 1 min interval. No smoothing is applied to this plot. b) Lag autocorrelation function shows the stability of the vectors.

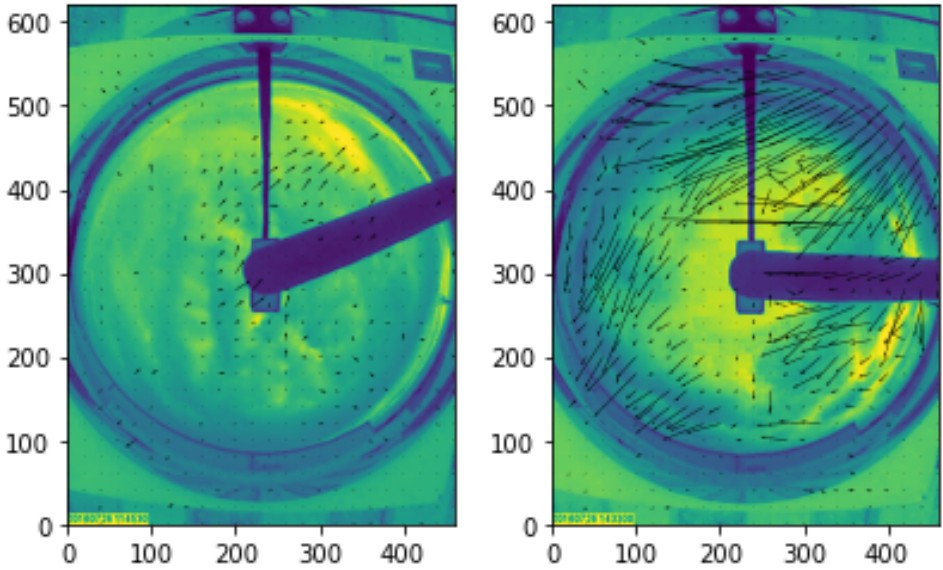

**Figure 13.** Two examples of dense cloud motion field using OF method, thinned by the factor of 20, show clustering of vectors in image space. Mean cloud motion in Figure 12a is underestimated due to the near-zero values.

tivity of the PC method to changes in block length, frame interval, and image resolution, as well as five combinations of the visible channels from a sky-viewing camera. These results are also applicable to satellite and radar-based motion estimation. Additionally, we compared the derived mean CMV from the PC method with the observed mean wind field from a collocated remote sensing instrument, and OF method. We also presented a method to detect raindrops on the rotating dome. However, the automated removal of contaminated images due to rain, snow, and other obscurities needs a more complex approach using advanced machine learning algorithms and labeled data.

The performance of different visible channels is comparable except for the red-to-blue channel ratio (RB). Although the RB is effective in segmenting clouds from the blue sky background (Dev et al., 2016), it smooths the cloud texture during overcast conditions, reducing the performance of the PC method. The red and grayscale performed slightly better than the blue and green channels. We find that larger block sizes provide a more stable estimation of cloud motion, and the stability benefits largely from the shortened interval between frames even for coarse-resolution camera data. Considering that the temporal changes in cloud patterns reduce the quality of the motion vectors, a shorter frame interval helps in maintaining the structure from one image to the next. However, a larger block size allows for a larger sample for stable correlation matching, achieving more stable estimates of the motion during disorganized cloud conditions (Fig. 8 c and d). Although averaging in time over the short frame interval is a better way to achieve reliable estimates, a higher sampling rate may not be always feasible. In these situations, *the large block size that can capture homogeneous motion is recommended* for block-based PC implementation. We also show that increasing the spatial resolution, i.e. increasing the number of pixels without decreasing the number of blocks, marginally affects the quality of the motion vectors. At the same time, reducing the frame interval from 60 sec to 30

sec outperforms quadrupling the resolution. Comparable results were obtained by Wang et al. (2018) for cloud segmentation using a ground-based camera.

Our analysis shows that doubling the frame rate outperforms quadrupling the resolution for PC. This non-intuitive result is very interesting in the context of edge computing. Because a shorter frame interval between the camera images effectively improves the quality of the CMVs, the application must have deterministic and low-latency access to sky images. Edge computing solves this problem efficiently by carefully placing and pairing computation with sensor data sources. Without incurring large data transfers and delays due to network outages, in an edge-computing platform like Sage, image data can be acquired and processed right next to the camera, in the field. The high-level motion estimation result which is much smaller and compresses efficiently can be communicated and archived for further studies.

The validation with constructed data and the comparison of PC and OF methods suggests that the quality of the motion vectors is sensitive to the error corrections and removal of the near-zero magnitudes in the post-processing. The dense OF field can be corrected using spatial clustering methods to produce valuable results. It is also possible to use the inputs from the cloud cover estimation plugin to correct the raw OF field. The issue of multi-layer clouds mentioned in Section 1 can be addressed using OF dense motion field using adaptive clustering as post-processing as opposed to adaptive blocks used in Peng et al. (2016). Further sensitivity and comparative studies with OF algorithm are needed to test this technique.

The distortion of the sky images near the horizon, due to the wide FOV of the fisheye lens, affects the accuracy of the mean cloud motion estimation. Therefore, the mean is estimated using the center portion of the images. The fisheye de-warping method can correct the regions near the horizon, where features are not heavily compressed.

## 5  Conclusion and Future Scope

Wind data retrieval from cloud motion vectors is an active area of research in satellite meteorology. Nevertheless, obtaining accurate wind retrievals from the ground-based optical camera images requires estimates of cloud-base heights, which is challenging without the Lidar-based methods. Moreover, despite assuming ideal CMV and cloud-base height estimates, the resulting winds may not align well with the observed cloud motion due to the substantial vertical extent of cumulus clouds and the influence of vertical wind shear on their motion. The growth and decay of clouds can also result in additional cloud motion components unrelated to the wind. Thermal infrared cameras can potentially help determine cloud-base heights and also cloud motion vectors for estimating winds in the future.

Current machine learning algorithms for automatic cloud identification underperform in the presence of thin clouds (Park et al., 2021). To this end, we are generating a dataset of thin clouds identified by scanning Mini Micro Pulse LiDAR (MiniMPL) and a co-located sky-viewing camera using an edge-computing paradigm. One of the objectives is to use the camera images to predict cloud boundaries and cloud motion and utilize the knowledge to adapt MiniMPL scan strategies in real time for optimal sampling in various environmental conditions. Thus, reducing the number of clear sky scans and targeting required clouds for the increased density of scans. Cloud locations predicted from CMV estimates can also be used in forecasting solar irradiance

in near real-time (Jiang et al., 2020; Radovan et al., 2021). The results of this study are helping to optimize image sampling and cloud motion estimation with edge-enabled camera systems.

*Code availability.* The Sage plugin implementation on the waggle platform is made available from https://portal.sagecontinuum.org/ and the full source code is available on GitHub at https://github.com/waggle-sensor/plugin-cmv-fftpc

.

*Data availability.* The data were obtained from the Atmospheric Radiation Measurement (ARM) user facility, a U.S. Department of Energy (DOE) Office of Science user facility managed by the Biological and Environmental Research Program. The ARM SGP data can be obtained from the following DOI 10.5439/1025309 (Total Sky Imager), 10.5439/1181954 (Ceilometer), and 10.5439/1025135 (Radar Wind Profiler). Sage camera data was collected at the Argonne Testbed for Multi-scale Atmospheric Observational Science (ATMOS). The ATMOS data used in this paper can be obtained by sending requests to the authors.

*Author contributions.* All authors contributed to the analysis plan and editing of the paper. BR designed the methodology and conducted sensitivity tests, data analysis, and plotting. SS, SP, and DD assisted in the algorithm selection and coding. SS, YK, and JS contributed to the development of the Sage plugin. RS, SP, NC, SS, and WG are responsible for the deployment, testing, and scheduling of the plugin on the waggle nodes for real-time use. BR, PM, and RJ lead the composition of the paper. PM contributed to the wind data analysis. SC, NF, and PB conceptualized the work and provided project oversight and direction.

*Competing interests.* The authors declare that they have no conflict of interest.

*Acknowledgements.* The Sage project is funded through the U.S. National Science Foundation's Mid-Scale Research Infrastructure program, NSF-OAC-1935984. The U.S. Department of Energy (DoE) Atmospheric Radiation Measurement (ARM) user facility supported the work under the field campaign AFC 07056 "ARMing the Edge: Demonstration of Edge Computing". Argonne National Laboratory's contribution is based upon work supported by Laboratory Directed Research and Development (LDRD) funding from Argonne National Laboratory, provided by the Director, Office of Science, of the U.S. Department of Energy under Contract No. DE-AC02-06CH11357. We gratefully acknowledge the computing resources provided on Bebop, a high-performance computing cluster operated by the Laboratory Computing Resource Center at Argonne National Laboratory. We thank the ATMOS observatory staff for their assistance. The algorithm, data analysis, and graphics were programmed in Python (https://www.python.org/) and the R programming language (http://www.R-project.org). We thank the two anonymous reviewers for their thoughtful comments and efforts that significantly improved the manuscript.

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
