# Peer review of "Optimizing cloud motion estimation on the edge with phase correlation and optical flow"

_Atmospheric Measurement Techniques, 2022_

## Author Comment (AC1)

**Author Response to the RC1: Anonymous Referee #1**

**General comments:**

The subject of the paper is relevant to the subject of this journal. The paper contains some new data relevant for publication. The presentation is clearly described, also the length of the paper is adequate. The title and the abstract are pertinent and understandable.

The authors give proper credit to related work and clearly indicate the own contribution.

>> We thank the reviewer for providing encouraging comments and valuable suggestions that improved the paper's readability. Our responses to specific comments are below.

**Specific Comments:**

- Section 2: the tools applied are not explained (e.g. which programming language, libraries for image processing, etc.)

>> We provided the Python source code of the Sage plugin in the code availability section. For readers' convenience, we have added the following statement in section 2.2:

> *The above implementation of the PC algorithm is available in several programming languages, notably C++, Python, and R in packages openCV (mulSpectrums), SkImage (phase_cross _correlation), and imagefx (pcorr3d) respectively. For this study, we used the Python implementation of Picel et al. (2018) using NumPy and SciPy packages. (See code availability section).*

- Section 2.1: an image of the used instrumentation (TSI, Sage camera) would be instructive.

>> We have added Figure 1 showing a TSI camera system and a Sage node with sky facing camera and other sensors.

[Figure]

a. Sage Node      b. Total Sky Imager

Also, explain "CEIL".

>> Thank you for pointing this out. We have now changed to using the full name for the ceilometer. The description of the working of the Ceilometer is also rearranged for clarity (See section 2.1.3). We hope this change makes the text easier to follow.

- Section 2.2: Explain in more detail the phase correlation and FFT methodology, not just refer to the paper of Leese et al.

C(mu, nu) is not explained.

>> The PC method is now described in more detail with added reference in section 2.2. The following paragraph is added after the PC method description for more clarity.

> Suppose image I2 is a spatially translated version of image I1. In that case, the phase covariance matrix Cov(p,q) is zero everywhere except for a sharp peak at the location corresponding to the displacement between the two images. The peak intensity is a good measure of the quality of the motion vector. Due to the reasons mentioned in Section 1, the actual peak in the covariance matrix can be fuzzy, and it corresponds to the best-fitting translational motion in the images. Sharp single-pixel peaks can sometimes occur in the covariance matrix, due to the high-frequency noise and artifacts in the images, which are flattened using Gaussian smoothing on Cov(p,q) with $\sigma = 3$.

C(mu, nu) is now explained in point 2 in section 2.2.

Section 3: The U and V components are used widely in the manuscript but nowhere explained in detail.

>> We thank the reviewer for suggesting this. The meaning of U and V components and their association with the X and Y image axis is now described at the end of section 2.2.

> For each image block, the peak covariance location is assigned as the local motion vector in image $I_2$ with respect to image $I_1$. As per the meteorological convention for winds, the U component is positive for an eastward flow, and the V component is positive for a northward flow. The location of the peak covariance from the center of the matrix gives the shift in the image features in the time period along the X and Y dimensions of the image. We saved X and Y shifts and computed the motion vectors per minute. The image top is oriented towards the north and therefore in the subsequent sections, the motion in the X and Y directions are referred to as U and V components, respectively.

---

## Author Comment (AC2)

**Response to the Comments of Reviewer #2**

Manuscript Ref: amt-2022-159

**Summary**

This paper by B. Raut, S. Collis, N. Ferrier, P. Muradyan, R. Sankaran, R. Jackson, S. Shahkarami, S. Park, D. Dematties, Y. Kim, J. Swantek, N. Conrad, W. Gerlach, S. Shemyakin, and P. Beckman discusses the retrieval of cloud motion vectors from distributed sensor systems (called sage nodes) equipped with a sky-facing camera. The paper discusses both total-sky imager (TSI) and Sage cameras. The idea is to use the phase-correlation (PC) method, which relies on fast Fourier transforms to obtain cloud displacements in predefined block cells, computing cross-correlations on successive images; a big advantage of this kind of method being that it is not computationally intensive. This very last point is strongly emphasised in the article, as the analysis is to be run on embedded computers. Discussions that follow are then meant to help decide how to optimise the retrieval algorithm to obtain the best results.

The original idea is good and sound, and it is also good that the paper presents field data that have been taken at the ARM research facility: "To validate the estimates of the CMV in our work, measurements from the co-located ceilometer and the wind profiling radar (...) were used from the SGP C1 site from October 14, 2017, to August 14, 2019." (lines 88-90).

We appreciate a thorough review of the paper. The major suggestions by the reviewer, which considerably improved the content of the paper,  will be incorporated into the revised manuscript.

The reviewer's main concerns are related to the wind data comparison (sec 3.3) and the validation of the algorithm. We are showing validation of the algorithm using synthetic data in this response. The data generated from existing images will also be shown in the paper. We hope the issues highlighted by the reviewer are properly addressed by this. Furthermore, we would like to clarify the following points in the context of the response.

1. The main objective of the paper is to study the sensitivity of the block-wise PC algorithm to the block size, time interval, color channel, and resolution of the images for better adaptation of the algorithm. Another important result is that it provides an auxiliary method to detect the raindrops on the mirror with the help of these motion vectors in a TSI camera, which should be applicable with any motion estimation algorithm.

2. As suggested by the reviewer, a separate section will be added for the validation of the PC method with synthetic data. We will also incorporate additional mathematical descriptions of the FFT-PC and NMF. This paper focuses on the better adaptation of the block-wise PC method which will guide many readers already using the method. Considering that both the optical flow and the PC methods have been used for several decades, the literature is also available regarding the validation, comparison, and limitations of the algorithms. Citations for the same are in the introduction and additional relevant citations will be added as well (See the reference section of the response).

3. Please note that wind retrieval is not an objective of the paper. In addition to accurate cloud motion vector (CMV) estimation, wind retrievals from the camera images also require good estimations of the cloud heights which itself has significant uncertainty without Lidar based approach. Despite assuming the perfect estimation of CMV and cloud-base height, the winds would not perfectly match the cloud motion as the cumulus clouds have significant depth and their motion is influenced by the wind shear in the entire depth of the cloud. In addition, the growth and decay of the clouds have an additional cloud motion component that is not related to the wind. Considering these factors wind retrieval is not in the scope of the current study.

4. We differ from the reviewer's opinion that correlations are not as quantitative as the metrics used by other studies (mainly RMSE). The correlation and the RMSE are related quantities and have one-to-one relations for standardized datasets over the same datasets (Barnston, 1992). Therefore, the correlations shown in this study accompanied by the significance test are a quantitative metric for accessing the errors in the CMV. Additionally, wind and CMV are not the same variables (as explained above) hence correlation is a suitable way to find the strength of association between them.

We will also clarify the above points in the text.

> While this sounds promising, as it stands this work remains far too qualitative, however. Notably, when comes the time for validation, which should be of utmost importance, lines 194-195 actually read "(...) this comparison may not be interpreted as a quantitative validation of the algorithm (...)".

Lines 194-195 refer to the wind data comparison and the reason for this is given in the same paragraph of the paper and also above in point 3.  In that section, we compared approximately 876 hours (selected for cloudiness) of valid wind data to show the long-term stability of the method. The uncertainties involved in such comparison are mentioned in the same paragraph of the paper. We have now clarified this in the following sentence.

*"Therefore, this comparison may not be interpreted as a quantitative validation of the algorithm **for wind retrievals**, however, significant correlations of the magnitudes indicate that the estimates of the instantaneous CMVs from the camera images are stable over a long period."*

> By the end of the paper, the reader is left wondering how good those retrievals actually are. The same remark holds regarding how one could meaningfully tune an algorithm that is never quantitatively compared to something else.
> More work and quantitative results are therefore needed before I can recommend its publication in Atmospheric Measurement Techniques.

We accept the reviewer's suggestion of adding a separate validation section in the paper. However, the validation part is not completely missing in the current version of the paper. We stated in section 3.2.1 that cloud motion should be stable on a minute-by-minute scale and hence random fluctuations in the CMV are due to the algorithm's errors. Zhen et al. also used the same assumption for validation and they counted large random fluctuations as an error. The lagged autocorrelation is a widely accepted method of quantifying randomness.  In our case, if the algorithm were to produce completely noisy values (i.e. zero skill) the correlation (in Fig 7) would have fallen to insignificant levels at lag-1.

Allowing that the FFT-PC algorithm is widely used in cloud motion estimation for decades, several other studies have reported the results (e.g. Leese et al. 1971, Schmetz et al. 1993; Kishtawal et al. 2009) as well as a comparison of the method (e.g. Zhen et al. 2019, Sawant et al. 2021 and references therein).

We are now providing preliminary validation in the response (See appendix) and we are also testing a synthetic dataset generated using a combination of additive noise, and color separation, followed by a known translation (This will be added in the revised paper). Please refer to Figure 1 below showing the early results of the synthetic data for 20-pix translation in 180x180 pixel image pairs. Our preliminary results are showing the expected distribution of the motion and also contamination due to spurious motion vectors peaking at zero. The peak around zero is also mentioned by Zhen et al 2019 as a failure of the PC method to detect motion. Removal of the spurious vectors requires post-processing depending on the application. Our method of choice for real-time correction is discussed in section 2.3 (Westerweel & Scarano, 2005). Thresholding of high-magnitude vectors and restricting near-zero values is necessary to improve the results.

[Figure]

Fig. 1: Distribution of the Motion vectors computed using FFT-PC for images translated by 20 pixels for 2000 images, not filtered for all clear sky images. These motion vectors are uncorrected.

**Response to Reviewer's comments**

> The main issue is that, beyond the mere existence of correlations, there must be more work to provide quantitative assessments in the validation (Sec. 3.3). How good are the retrieved wind components U and V? This must be clearly communicated. It cannot just end on a quick qualitative note. The strategy to derive trustworthy winds from this should be spelled out.

We understand that our comparison with the wind vectors caused a misunderstanding. As noted before (in point 3) the study does not attempt to retrieve the wind vectors and it will be clarified in the text.
Section 3.3 'Comparison with Wind Data' uses 551 days of wind data with 876 cloudy hours with stable cloud-base height identified from the ceilometer. Using such data we extract the winds at the appropriate heights for CMV comparison and presented correlations, their significance level, and the heightwise slope of the relationship. This is more than a quick qualitative note. We realized that the use of the word qualitative mislead the reader's interpretation hence we will to remove it.

> Other similar peer-reviewed papers are not satisfied with autocorrelations and a qualitative comparison. They do compare against other methods, use synthetic datasets, use quantitative metrics, ...

A new subsection (validation using synthetic data) will be added to alleviate this concern of the reviewer. However, it should be noted that the major objective of the other papers mentioned by the reviewer is to compare the methods. The focus of our paper is on presenting the sensitivity study for the PC method.

> Figure 9 is not really readable and not convincing. The units for both the x and y axes must be displayed. Moreover, in the text the attention of the reader should be brought to the fact that the ranges spanned in x and y are widely different, which can be misleading.

We thank the reviewer for this suggestion. We will mention the units and ranges in the revised version.

> The accompanying discussion should address the issues that appear in Figure 9, e.g. if all was perfect, should these be comparable (i.e. essentially follow a y = x line)? If yes, then it should be discussed.

The wind vectors and CMV are not exactly the same variables and they do not match one-to-one even in the ideal situation. This is due to the factors mentioned above in point 3. Therefore, we provided a cautionary statement to the effect that this is not a validation of wind retrieval. However, *winds and CMV should correlate*. We will clarify it in the text.

> The rain points are deemed problematic, but the remaining points do not seem much better.

We agree with the reviewer's concern. We could only remove rain points using the rotation of the vectors. However, completely snow-covered images can not be removed with this method as the rotation is not seen under the snow. Despite using stable cloud-based data from the ceilometer, clear sky periods have contaminated the data during which cloud motions are near zero. The discussion of this aspect will be added in the text and we will replot the figure to check if additional quality control improves the results near the origin. It should be noted that this comparison is made with a 30-sec image interval for approximately 500 days that involves several types of weather conditions and real-life issues such as birds and dirt on the mirror which can not be removed with the current method. Considering the above contaminations the shown correlations are encouraging. We will emphasize on the contamination issue in the revised manuscript.

> More quantitative information must also be given and discussed together with the figure, for instance in the form of root-mean-squared-differences and biases, applied to the difference between the value obtained from the algorithm and the expected one used for validation. This should come with a discussion, especially if the discrepancies are large.

We agree with the reviewer that RMSE and bias are better comparisons for the same variables. However, as mentioned before the wind vectors comparison with the CMV is not the same variable comparison. The purpose of the wind comparison is to show that the CMV is well-correlated with the wind suggesting that the algorithm is stable over varied situations over a long time and not just good in specific cloudy conditions. As referred to in point 4 above, the correlations are quantitatively related to RMSE and in addition, the slopes suggest the biases.

> Several other peer-reviewed works similarly dedicated to cloud motion, sky-looking cameras, TSI, and using various techniques (both block-matching and optical flow; both from the images themselves, or going to Fourier space) do compare their results against other established methods. Those include for instance Zhen et al. (2019) (which is cited in the current paper) and Peng et al. (2016). Depending on the approach, a number of evaluation metrics are also given e.g. in Peng et al. (2016). Something along such lines is possible for the current paper. Such a comparison could be done on a more powerful machine, since the idea is to validate. To be recommended for publication, the paper should at least clearly show that the retrievals make sense.
>
> Alternatively, they could also use synthetic datasets, once again as done in other similar works such as Zhen et al. (2019) and Peng et al. (2016).

Thank you for making this suggestion. We have presented the validation results above which have improved the manuscript. We did not compare the results with the other methods as the main objective of the paper is to show the sensitivity of the block sizes, interval, color, and resolution of the images to the PC algorithm. We hope that our response to the earlier suggestions clarifies this issue.

**Response to further key issues and comments:**

> 1. The paper should explicitly discuss the well-known issue of multilayer clouds, which is currently not mentioned though it is discussed among other problems in Leese et al. (1971) and Zhen et al. (2019), which are both cited as PC-method references in the current paper (lines 19-20).

We thank the reviewer for bringing up this point. This is an interesting issue that the tracking algorithms have to deal with. The introduction section of the paper referred to the clouds moving in different directions (multilayer). Therefore the block-wise algorithm is suggested to mitigate the issue. We will further discuss this in the introduction section.

> -- This is from Zhen et al. (2019) (their page 2): "However, as shown in the prior work, FPCT [Fourier phase correlation theory] is unable to recognize multiple motion displacement vectors from different cloud layers because it can merely extract one displacement value for a global image [46].

This is true when FFTPC is used over the entire image (as in Zhen et al) and this issue can be effectively mitigated using the blockwise method which is discussed in this paper.

> Interferences such as sky background effect, pixel superposition, the motion of multiple cloud layers, and irregular cloud deformation can all cause random noise leading to significant displacement calculation errors.".

We agree with the reviewer. These issues are true for any cloud tracking algorithm. Such errors usually appear as fluctuations from the surrounding motion vectors and can be removed by various methods. Fig 10 in Zhen et al shows that the performance of the pure FFTPC on global images is as good as other preprocessing methods but with not considering the outliers that can be easily removed, except for the optical flow underperforming FFTPC. Their Fig 11 also shows outliers in transformed CMVs which are cleared thanks to the ensemble method.

We used Westerweel & Scarano (2005), the details of which will be added in the revised version.

> -- Zhen et al. (2019) also add, on their page 3: "According to the algorithm principle of FPCT, the cloud displacement calculation result is either correct or unacceptable. The probability for correct result depends on the noise intensity." Note that they then move on and discuss the "low robustness of the FPCT method".

Thank you for quoting this from Zhen et al. We have seen similar behavior (See above figure 1). This is now discussed in the synthetic data section.

However, a direct comparison between the methodology of Zhen et al with our work is not justified as they used a different approach. Zhen's methodology differs from this paper in some key aspects. They have an ensemble approach. Therefore, several methods and their variations by changing parameters are used to get the ensemble of predictions and the mean of the Gaussian distribution fitted to the ensemble is the best estimate. Note that the similarity indices are used in assessing the quality of the transformed images, not for the quality of the vectors.

Our approach, on the other hand, is to use the blockwise estimation of the motion which gives multiple vectors and then remove the outliers before taking the mean.

Zhen et al., also computed global shift for the entire image which only works best in case of the homogeneous motion over the entire domain, and the block-wise PC method is always recommended (discussed in sec 1). The unreliable results from the PC method in Zhen et al can be attributed to this assumption of homogeneity of the motion. Our experience with global shift estimation is similar to Zhen et al. Moreover, the frequency of extremely large CMD values in their paper can also be due to the noisy CC matrix, where the peak is found at the boundaries, we also encountered this but less frequently than Zhen et al., due to the blocks-wise implementation. This was the reason to use Vmax to remove the outliers. See sec.

> The second quote moreover emphasises the issue of not performing preprocessing in the current paper and seems to contradict the claim that "The PC method can be implemented without preprocessing images" (lines 36-37). Again, I do stress that Zhen et al. (2019) is a reference of the current paper for the PC method.

We tried preprocessing using openCV background removal and found no improvement in the quality during the development process. This is due to the independence of the phase shift to the lighting conditions and the Gaussian smoothing of the correlation matrix. The position of the peaks in the correlation matrix is related to the changes in the larger textures and not to the change in magnitudes. We have now added Zhen et al as a reference to the FFT-PC method as it also shows a comparison with the other methods. However, they are not using the blockwise algorithm.

2. The authors should also add some discussion in the paper on the issue of image distortion and how it can quantitatively affect their retrieved winds. Indeed, for TSI (also used in this work), "image distortion compromises the accuracy of the detected motion vectors, especially around the boundary of an image" (Peng et al. (2016)). With the PC method, the distortion issue does not vanish once we go to Fourier space. From the images and animations from the current paper, no filtering seems to be performed on the edges though (in fact, arrows are even derived for the TSI supporting arm).

We couldn't agree more with the reviewer's concerns. The distortion at the boundaries can not be simply removed by regridding the hemispheric data to a square image. The boundary of the images is excluded from the mean calculation for the same reason.

3. The need for a not-too-heavy algorithm is emphasised: e.g. 'computational overhead complicating their use in real-time applications' (line 23), 'computational efficiency of the algorithm is critical' (line 36), '[FFT] is computationally efficient, and hence a natural choice' (line 40). However, it would seem that the objective should at least be to obtain usable retrieved winds (a too simplistic algorithm could easily lead to poor retrievals, as already stressed). While the authors do not give the technical specifications of the sage nodes in the paper (e.g. CPU number of cores and clock speeds; RAM), the Sage website actually reads "Cyberinfrastructure for AI at the Edge" and the associated proceedings, Beckman et al. (2016), which is both cited in and shares authors with the current paper, talks about computer vision (5 times) and OpenCV (2 times), which are quite heavy tools. Therefore, following on the preceding remarks, it seems likely that a more sophisticated algorithm that would at least properly take into account the caveats commonly discussed and addressed in earlier peer-reviewed works on the subject might be both needed to obtain satisfactory results, while being achievable in practice. Has this been attempted, and if not why?

We agree with the reviewer. However, the Sage nodes run many applications using OpenCV and deep learning models, some of them are critical, for example, traffic estimators in the city and wildfire detection in the forests. These applications take the bulk of the processing powers due to the deep learning models. Therefore, it is important to try for low processing for most applications. In the future, we may use more complicated algorithms by adapting an advanced machine learning approach to estimate the cloud motion after accessing their value addition to the final product such as solar irradiance estimators.

4. Compared to similar peer-reviewed works, the paper is bit light on the mathematical front regarding the techniques employed. It would be good for the unfamiliar reader if the article was more self-contained.

Thank you for suggesting this. We did not give the mathematical description of the PC method or median fluctuation method in the paper as it is available in the referred papers. However, we will add it in the revised version.

> 5. Note: The size in megabytes of Figure 2 in the paper should absolutely be reduced. In the pdf, page 6 alone is indeed responsible for more than 20 MB in the final document file size.

Done. Thank you so much for bringing this to our attention.

> 6. For the references, it would be preferable to also include a peer-reviewed work whenever it makes sense and is possible, and not only proceedings as is currently the case for optical flow.

Thank you for the suggestion. We will review the journal publications on optical flow relevant to cloud motion and cite them appropriately.

**References:**

- Barnston, A. G. (1992). Correspondence among the correlation, RMSE, and Heidke forecast verification measures; refinement of the Heidke score. Weather and Forecasting, 7(4), 699-709.

- Kishtawal, C., S. Deb, P. Pal, and P. Joshi, 2009: Estimation of atmospheric motion vectors from Kalpana-1 imagers. J. Appl. Meteor. Climatol., 48, 2410–2421.

- Leese, J. A., C. S. Novak, and B. B. Clark, 1971: An automated technique for obtaining cloud motion from geosynchronous satellite data using cross correlation. J. Appl. Meteor., 10, 118–132

- Liu, T., Merat, A., Makhmalbaf, M. H. M., Fajardo, C., & Merati, P. (2015). Comparison between optical flow and cross-correlation methods for extraction of velocity fields from particle images. Experiments in Fluids, 56(8), 1-23.

- Sawant, M., Shende, M. K., Feijóo-Lorenzo, A. E., & Bokde, N. D. (2021). The State-of-the-Art Progress in Cloud Detection, Identification, and Tracking Approaches: A Systematic Review. Energies, 14(23), 8119.

- Schmetz, J., K. Holmlund, J. Hoffman, B. Strauss, B. Mason, V. Gaertner, A. Koch, and L. Van De Berg, 1993: Operational cloud-motion winds from Meteosat infrared images. J. Appl. Meteor., 32, 1206–1225.

- Westerweel, J., & Scarano, F. (2005). Universal outlier detection for PIV data. Experiments in fluids, 39(6), 1096-1100.

- Zhen, Z., Xuan, Z., Wang, F., Sun, R., Duić, N., & Jin, T. (2019). Image phase shift invariance based multi-transform-fusion method for cloud motion displacement calculation using sky images. Energy Conversion and Management, 197, 111853.

**Appendix: Testing with Synthetic Data**

To test this, make two dummy images with uniform noise and add an object with irregular structures for tracking.

**Tracking a Single Rigid Object**

[Figure]

- shift from noisy crossCov is (6, 1)

- shift from smooth crossCov is (6, 1)

- shift in this image is (5, 0)

**Tracking a Changing Object**

[Figure]

- shift from noisy crossCov is (2, -4)

- shift from smooth crossCov is (3, -5)

- shift in this image is (2, -5)

**Tracking Multiple Objects with Incoherent Motion**

[Figure]

- shift from noisy crossCov is (2, -5)

- shift from smooth crossCov is (3, -4)

- real shift: object1=(2, -5) and object2=(1, -3)

- Average shift is (1, -4)

**Tracking Multiple Objects with new Object**

[Figure]

- shift from noisy crossCov is (2, -4)

- shift from smooth crossCov is (3, -4)

- real shift: object1=(2, -5) and object2=(1, -3)

- Average shift is (1, -4)

**Real cloud element significantly changed between the images after long interval.**

[Figure]

- shift from noisy crossCov is (-6, 0)

- shift from smooth crossCov is (-2, 2)

- Manually guessed shift is approx. (-2, 0).

---

## Author Response (AR1)

**Final Response to reviewers' comments**

`Ref: AMT-2022-159`

**Author's response**

We thank both reviewers for their valuable suggestions, which significantly improved the manuscript. The revision incorporates all the suggestions of reviewers 1 and 2 as mentioned in our point-by-point responses (AC1 and AC2) submitted in the discussion phase.

The revised paper addresses two major concerns of reviewer #2.

1. Quantitative validation of the method with synthetic data (Added sections 2.3 and 3.1). and
2. Comparison of motion estimated using phase correlation and optical flow methods (Added sections 2.7 and 3.5).

In this revision, the following major changes are made to the manuscript to incorporate the reviewer's suggestions.

1. Instrument photo is added as suggested by reviewer #1 (Fig. 1).
2. Constructed data section 2.3: detailing the steps for generating validation data.
3. Optical Flow section 2.7: Brief description of optical and the specifications of the method presented in this paper.
4. Validation section 3.1: contains a new validation figure (Fig. 3) and a quantitative table showing errors (Table 1).
5. Comparison with optical flow section 3.5: contains two new figures (Fig 12 and 13) comparing PC and optical flow methods and showing the effect of postprocessing.
6. The two figures from the appendix showing examples of raw motion vectors are removed. Figure 12a is now showing an example of the raw motion vectors for PC and OF both.
7. Figure 9 (Now Figure 11) is updated as per reviewers' suggestions.
8. Sections 1, section 2.2, and section 4 are significantly modified with the new discussions, references, and added mathematical details. (See attached diff.pdf)
9. Mathematical details of NMF methods as well as PC and OF methods are provided (Sec 2.2, 2.4, and 2.7).
10. Explanations of U and V components as well as software package information are given as suggested by reviewer 1 (Section 2.2).
11. The title of the paper has changed to reflect the new additions. "Cloud motion on the edge with phase correlation and optical flow"
12. We have also incorporated all the minor suggestions of the reviewers as mentioned in AC1 and AC2.

We hope that the revised version of the paper is suitable for publication in Atmos. Meas. Tech.

**Reviewer 1**

**General comments:**

The subject of the paper is relevant to the subject of this journal. The paper contains some new data relevant for publication. The presentation is clearly described, also the length of the paper is adequate. The title and the abstract are pertinent and understandable.
The authors give proper credit to related work and clearly indicate the own contribution.

We thank the reviewer for providing encouraging comments and valuable suggestions that improved the paper's readability. Our responses to specific comments are below.

**Specific Comments:**

- Section 2: the tools applied are not explained (e.g. which programming language, libraries for image processing, etc.)

We provided the Python source code of the Sage plugin in the code availability section. For readers' convenience, we have added the following statement in section 2.2:
The above implementation of the PC algorithm is available in several programming languages, notably C++, Python, and R in packages openCV (mulSpectrums), SkImage (phase_cross _correlation), and imagefx (pcorr3d) respectively. For this study, we used the Python implementation of Picel et al. (2018) using NumPy and SciPy packages. (See code availability section).

- Section 2.1: an image of the used instrumentation (TSI, Sage camera) would be instructive.

We have added Figure 1 showing a TSI camera system and a Sage node with sky facing camera and other sensors.

Also, explain "CEIL".

Thank you for pointing this out. We have now changed to using the full name for the ceilometer. The description of the working of the Ceilometer is also rearranged for clarity (See section 2.1.3). We hope this change makes the text easier to follow.

- Section 2.2: Explain in more detail the phase correlation and FFT methodology, not just refer to the paper of Leese et al.
  C(mu, nu) is not explained.

The PC method is now described in more detail with added reference in section 2.2.

Section 3: The U and V components are used widely in the manuscript but nowhere explained in detail.

We thank the reviewer for suggesting this. The meaning of U and V components and their association with the X and Y image axis is now described at the end of section 2.2.

**Reviewer 2**

**Summary**

> This paper by B. Raut, S. Collis, N. Ferrier, P. Muradyan, R. Sankaran, R. Jackson, S. Shahkarami, S. Park, D. Dematties, Y. Kim, J. Swantek, N. Conrad, W. Gerlach, S. Shemyakin, and P. Beckman discusses the retrieval of cloud motion vectors from distributed sensor systems (called sage nodes) equipped with a sky-facing camera. The paper discusses both total-sky imager (TSI) and Sage cameras. The idea is to use the phase-correlation (PC) method, which relies on fast Fourier transforms to obtain cloud displacements in predefined block cells, computing cross-correlations on successive images; a big advantage of this kind of method being that it is not computationally intensive. This very last point is strongly emphasised in the article, as the analysis is to be run on embedded computers. Discussions that follow are then meant to help decide how to optimise the retrieval algorithm to obtain the best results.
> The original idea is good and sound, and it is also good that the paper presents field data that have been taken at the ARM research facility: "To validate the estimates of the CMV in our work, measurements from the co-located ceilometer and the wind profiling radar (...) were used from the SGP C1 site from October 14, 2017, to August 14, 2019." (lines 88-90).

We appreciate a thorough review of the paper. The major suggestions by the reviewer are incorporated into the revised manuscript which considerably improved the content of the paper.

The reviewer's main concerns are related to the wind data comparison (sec 3.3) and the validation of the algorithm. We are showing validation of the algorithm using synthetic data in Section 3.1. We have also added the comparison with the optical flow method in section 3.5. We hope the issues highlighted by the reviewer are properly addressed by this.

> While this sounds promising, as it stands this work remains far too qualitative, however. Notably, when comes the time for validation, which should be of utmost importance, lines 194-195 actually read "(...) this comparison may not be interpreted as a quantitative validation of the algorithm (...)".

Lines 194-195 refer to the wind data comparison and the reason for this is given in the same paragraph of the paper and also above in point 3. In that section, we compared approximately 876 hours (selected for cloudiness) of valid wind data to show the long-term stability of the method. The uncertainties involved in such comparison are mentioned in the same paragraph of the paper. We have now clarified this in the following sentence.

*"Therefore, this comparison may not be interpreted as a quantitative validation of the algorithm **for wind retrievals**, however, significant correlations of the magnitudes indicate that the estimates of the instantaneous CMVs from the camera images are stable over a long period."*

> By the end of the paper, the reader is left wondering how good those retrievals actually are. The same remark holds regarding how one could meaningfully tune an algorithm that is never quantitatively compared to something else.
> More work and quantitative results are therefore needed before I can recommend its publication in Atmospheric Measurement Techniques.

We accept the reviewer's suggestion of adding a separate validation section in the paper.
However, the validation part is not completely missing in the current version of the paper. We stated in

section 3.2.1 that cloud motion should be stable on a minute-by-minute scale and hence random fluctuations in the CMV are due to the algorithm's errors. Zhen et al. also used the same assumption for validation and they counted large random fluctuations as an error. The lagged autocorrelation is a widely accepted method of quantifying randomness.  In our case, if the algorithm were to produce completely noisy values (i.e. zero skill) the correlation (in then Fig 7 now Fig 8) would have fallen to insignificant levels at lag-1.

Allowing that the FFT-PC algorithm is widely used in cloud motion estimation for decades, several other studies have reported the results (e.g. Leese et al. 1971, Schmetz et al. 1993; Kishtawal et al. 2009) as well as a comparison of the method (e.g. Zhen et al. 2019, Sawant et al. 2021 and references therein).

_We have provided detailed validation in two parts in earlier the response (See AC2) and now added validation in Section 3.1 as well as comparison with optical flow in section 3.5.

**Response to Reviewer's comments**

> The main issue is that, beyond the mere existence of correlations, there must be more work to provide quantitative assessments in the validation (Sec. 3.3). How good are the retrieved wind components U and V? This must be clearly communicated. It cannot just end on a quick qualitative note. The strategy to derive trustworthy winds from this should be spelled out.
> Other similar peer-reviewed papers are not satisfied with autocorrelations and a qualitative comparison. They do compare against other methods, use synthetic datasets, use quantitative metrics, ...

Detail response to this comment has been provided in the author's comments AC2. We have now added a validation section 3.1 with a table to the revised version to alleviate this concern of the reviewer.

> Figure 9 is not really readable and not convincing. The units for both the x and y axes must be displayed. Moreover, in the text the attention of the reader should be brought to the fact that the ranges spanned in x and y are widely different, which can be misleading.

Done. See Figure 11 and the text in section 3.4.

> The accompanying discussion should address the issues that appear in Figure 9, e.g. if all was perfect, should these be comparable (i.e. essentially follow a y = x line)? If yes, then it should be discussed.

Detail explanation of this has been provided in the author's comments AC2. We have now clarified this in section 3.4.

> The rain points are deemed problematic, but the remaining points do not seem much better.

Detail explanation of this has been provided in the author's comments AC2. We have further clarified this in section 3.4.

> More quantitative information must also be given and discussed together with the figure, for instance in the form of root-mean-squared-differences and biases, applied to the difference between the value obtained from the algorithm and the expected one used for validation. This should come with a discussion, especially if the discrepancies are large.

Table 1 is added to address this concern.

> Several other peer-reviewed works similarly dedicated to cloud motion, sky-looking cameras, TSI, and using various techniques (both block-matching and optical flow; both from the images themselves, or going to Fourier space) do compare their results against other established methods. Those include for instance Zhen et al. (2019) (which is cited in the current paper) and Peng et al. (2016). Depending on the approach, a number of evaluation metrics are also given e.g. in Peng et al. (2016). Something along such lines is possible for the current paper. Such a comparison could be done on a more powerful machine, since the idea is to validate. To be recommended for publication, the paper should at least clearly show that the retrievals make sense.
> Alternatively, they could also use synthetic datasets, once again as done in other similar works such as Zhen et al. (2019) and Peng et al. (2016).

Thank you for making this suggestion. We have presented the validation results in section 3.1 and optical flow results in section 3.5, which have improved the manuscript. We hope that our response to the earlier suggestions clarifies this issue.

**Response to further key issues and comments:**

> 1. The paper should explicitly discuss the well-known issue of multilayer clouds, which is currently not mentioned though it is discussed among other problems in Leese et al. (1971) and Zhen et al. (2019), which are both cited as PC-method references in the current paper (lines 19-20).

We thank the reviewer for bringing up this point. This is an interesting issue that the tracking algorithms have to deal with. The introduction section of the paper referred to the clouds moving in different directions (multilayer). Therefore the block-wise algorithm is suggested to mitigate the issue. _We have added a reference to Peng et al in the introduction section and explicitly mentioned the multilayer clouds.

> -- This is from Zhen et al. (2019) (their page 2):  "However, as shown in the prior work, FPCT [Fourier phase correlation theory] is unable to recognize multiple motion displacement vectors from different cloud layers because it can merely extract one displacement value for a global image [46].

This is true when FFTPC is used over the entire image (as in Zhen et al) and this issue can be effectively mitigated using the blockwise method which is discussed in this paper.

> Interferences such as sky background effect, pixel superposition, the motion of multiple cloud layers, and irregular cloud deformation can all cause random noise leading to significant displacement calculation errors.".

We agree with the reviewer. These issues are true for any cloud tracking algorithm. Such errors usually appear as fluctuations from the surrounding motion vectors and can be removed by various methods. Fig 10 in Zhen et al shows that the performance of the pure FFTPC on global images is as good as other preprocessing methods but with not consider the outliers that can be easily removed, except for the optical flow underperforming FFTPC. Their Fig 11 also shows outliers in transformed CMVs which are cleared thanks to the ensemble method.

_We used Westerweel & Scarano (2005), the details of which are now added in the revised version.

> -- Zhen et al. (2019) also add, on their page 3: "According to the algorithm principle of FPCT, the cloud displacement calculation result is either correct or unacceptable. The probability for correct

> result depends on the noise intensity." Note that they then move on and discuss the "low robustness of the FPCT method".

Thank you for quoting this from Zhen et al. We have seen similar behavior (See Figure 3). *This is now discussed in the validation section 3.1*. Please, check AC2 for a detail response.

> The second quote moreover emphasises the issue of not performing preprocessing in the current paper and seems to contradict the claim that "The PC method can be implemented without preprocessing images" (lines 36-37). Again, I do stress that Zhen et al. (2019) is a reference of the current paper for the PC method.

A detailed response to this comment is provided in AC2. We have now shown the comparison of optical flow and PC methods in Section 3.5.

> 2. The authors should also add some discussion in the paper on the issue of image distortion and how it can quantitatively affect their retrieved winds. Indeed, for TSI (also used in this work), "image distortion compromises the accuracy of the detected motion vectors, especially around the boundary of an image" (Peng et al. (2016)). With the PC method, the distortion issue does not vanish once we go to Fourier space. From the images and animations from the current paper, no filtering seems to be performed on the edges though (in fact, arrows are even derived for the TSI supporting arm).

We couldn't agree more with the reviewer's concerns. The distortion at the boundaries can not be simply removed by regrinding the hemispheric data to a square image. A more calibrated de-warping method may be helpful. In our case, the boundary of the images is excluded from the mean calculation for the same reason. The discussion is now added in section 4.

> 3. The need for a not-too-heavy algorithm is emphasised: e.g. 'computational overhead complicating their use in real-time applications' (line 23), 'computational efficiency of the algorithm is critical' (line 36), '[FFT] is computationally efficient, and hence a natural choice' (line 40). However, it would seem that the objective should at least be to obtain usable retrieved winds (a too simplistic algorithm could easily lead to poor retrievals, as already stressed). While the authors do not give the technical specifications of the sage nodes in the paper (e.g. CPU number of cores and clock speeds; RAM), the Sage website actually reads "Cyberinfrastructure for AI at the Edge" and the associated proceedings, Beckman et al. (2016), which is both cited in and shares authors with the current paper, talks about computer vision (5 times) and OpenCV (2 times), which are quite heavy tools. Therefore, following on the preceding remarks, it seems likely that a more sophisticated algorithm that would at least properly take into account the caveats commonly discussed and addressed in earlier peer-reviewed works on the subject might be both needed to obtain satisfactory results, while being achievable in practice. Has this been attempted, and if not why?

We agree with the reviewer. However, the Sage nodes run many applications using OpenCV and deep learning models, some of them are critical, for example, traffic estimators in the city and wildfire detection in the forests. These applications take the bulk of the processing powers due to the deep learning models. Therefore, it is important to try for low processing for most applications. In the future, we may use more complicated algorithms by adapting an advanced machine learning approach to estimate the cloud motion after accessing their value addition to the final product such as solar irradiance estimators.

4. Compared to similar peer-reviewed works, the paper is bit light on the mathematical front regarding the techniques employed. It would be good for the unfamiliar reader if the article was more self-contained.

Thank you for suggesting this. We have now added it in the revised version in sections 2.2, 2.3, 2.4 and 2.7.

5. Note: The size in megabytes of Figure 2 in the paper should absolutely be reduced. In the pdf, page 6 alone is indeed responsible for more than 20 MB in the final document file size.

Done. Thank you so much for bringing this to our attention.

6. For the references, it would be preferable to also include a peer-reviewed work whenever it makes sense and is possible, and not only proceedings as is currently the case for optical flow.

Thank you for the suggestion. References for Apke et al., 2022; Mondragón et al., 2020; Peng et al., 2016 are added in section 1.

**References:**

- Apke, J. M., Noh, Y.-J., and Bedka, K.: Comparison of Optical Flow Derivation Techniques for Retrieving Tropospheric Winds from Satellite Image Sequences, J Atmos Ocean Tech, 2022.
- Barnston, A. G. (1992). Correspondence among the correlation, RMSE, and Heidke forecast verification measures; refinement of the Heidke score. Weather and Forecasting, 7(4), 699-709.
- Kishtawal, C., S. Deb, P. Pal, and P. Joshi, 2009: Estimation of atmospheric motion vectors from Kalpana-1 imagers. J. Appl. Meteor. Climatol., 48, 2410–2421.
- Leese, J. A., C. S. Novak, and B. B. Clark, 1971: An automated technique for obtaining cloud motion from geosynchronous satellite data using cross correlation. J. Appl. Meteor., 10, 118–132
- Liu, T., Merat, A., Makhmalbaf, M. H. M., Fajardo, C., & Merati, P. (2015). Comparison between optical flow and cross-correlation methods for extraction of velocity fields from particle images. Experiments in Fluids, 56(8), 1-23.
- Mondragón, R., Alonso-Montesinos, J., Riveros-Rosas, D., and Bonifaz, R.: Determination of cloud motion applying the Lucas-Kanade method to sky cam imagery, Remote Sensing, 12, 2643, 2020.
- Peng, Z., Yu, D., Huang, D., Heiser, J., and Kalb, P.: A hybrid approach to estimate the complex motions of clouds in sky images, Sol Energy, 138, 10–25, 2016
- Sawant, M., Shende, M. K., Feijóo-Lorenzo, A. E., & Bokde, N. D. (2021). The State-of-the-Art Progress in Cloud Detection, Identification, and Tracking Approaches: A Systematic Review. Energies, 14(23), 8119.
- Schmetz, J., K. Holmlund, J. Hoffman, B. Strauss, B. Mason, V. Gaertner, A. Koch, and L. Van De Berg, 1993: Operational cloud-motion winds from Meteosat infrared images. J. Appl. Meteor., 32, 1206–1225.
- Westerweel, J., & Scarano, F. (2005). Universal outlier detection for PIV data. Experiments in fluids, 39(6), 1096-1100.
- Zhen, Z., Xuan, Z., Wang, F., Sun, R., Duić, N., & Jin, T. (2019). Image phase shift invariance based multi-transform-fusion method for cloud motion displacement calculation using sky images. Energy Conversion and Management, 197, 111853.

---

## Author Response (AR2)

**Final Response to Associate Editor Decision**

Manuscript Ref.: amt-2022-159
*Cloud motion on the edge with phase correlation and optical flow* by Bhupendra A. Raut et al.

**Reviewer #2**

> One promising aspect of this paper is that wind-profiling radar and ceilometer measurements were taken "to validate the estimates of the CMV" (cf. Sec. 2.1.3. of the current version). However, the authors have now indicated in their answer to my requests for a more qualitative validation that retrieving the wind is actually not an objective of the paper/algorithm. As a result, the connection between these wind-profiler-radar and ceilometer atmospheric measurements and the optimisation of their CMV algorithm now becomes quite unclear. Some things have certainly been much improved in the paper as it stands now, such as welcome added details about the methods, and the use of synthetic data, which now give something to compare against (in terms of shifts in pixels). But it might be more of an image-motion detection technique paper, than an atmospheric-measurement-technique one. It is not clear what reference empirical atmospheric measurements are actually used to compare against and therefore how meaningful the assessment of how good the retrievals and the optimisation of the algorithm are. This, especially given what follows.

> One point is that there are already many papers that are using more sophisticated algorithms with careful preprocessing to avoid well-known issues/pitfalls leading to unsatisfactory results (this was also an essential part of the original report). When asked why those known approaches to obtain better results were not implemented here (pointing out that the Sage nodes are able to run ML and OpenCV), the answer has been "the Sage nodes run many applications using OpenCV and deep learning models, some of them are critical, for example, traffic estimators in the city and wildfire detection in the forests. These applications take the bulk of the processing powers due to the deep learning models. Therefore, it is important to try for low processing for most applications. In the future, we may use more complicated algorithms by adapting an advanced machine learning approach to estimate the cloud motion after accessing their value addition to the final product such as solar irradiance estimators." Because of this conscious choice, what this paper actually brings to the literature is then limited; this might be an acceptable justification for a proceedings, but probably less so for an article in a journal. At the very least, the limited impact of not addressing these issues should be demonstrated.

We appreciate the reviewer's efforts to evaluate the paper again. We understand their concerns regarding the wind retrieval from the CMV and, as mentioned in the earlier response, wind retrieval needs the height estimations. For this, we are considering using thermal sensors in the future to derive the height of the cloud bases. However, the discussion of that is outside the scope of the current paper. The focus of the current paper is *to test the sensitivity of the phase correlation (PC) algorithm and compare it to the optical flow (OF) method*. The paper's contribution is in providing

insights into the sensitivity of the block-wise PC method, which is not available in the peer-reviewed literature to our knowledge.

The wind and ceilometer measurements served as additional validation, showing consistency with independent atmospheric measurements over two years period. Such comparison with *a long-term dataset* is also not found in the literature and it was made possible thanks to the ARM SGP user facility. The application of CMV (alone) for targeted miniMPL scanning is valuable for cloud and aerosol research (Mentioned in Section 5).

Moreover, despite the recent advent of sophisticated methods, especially in the AI/ML domain, traditional computer vision methods (e.g. PC, OF, Kalman filtering) will remain in use due to their flexibility, efficiency, and more so their explainability. Therefore, the results of this paper will be of interest to researchers who are using PC/OF methods.

We have now clarified our objectives (in the Introduction section and modified the title) in the final version of the paper. We are very grateful for the reviewer's earlier suggestions that substantially improved the contents of the manuscript.

**Associate Editor decision:**

> **Publish subject to minor revisions (review by editor) by Ad Stoffelen**
> The 2 reviews are rather different, but both of interest. The technical aspects of atmospheric cloud motion measurement are now much improved and it is clear that the manuscript focuses on these aspects, which satisfies the first reviewer. The abstract and title also focus on these aspects. Nevertheless, the second reviewer is looking for the practical usefulness of the measurements, which is less clear indeed and of course very relevant. I recommend to sincerely consider the reviewer's comments with the aim to further clarify the manuscript in terms of these latter aspects in a minor revision.

We thank Associate Editor Ad Stoffelen for the opportunity to revise our manuscript and appreciate Prof. Robert Höller and anonymous reviewer #2 for their efforts in providing their perspectives.
We understand that the two reviews present differing views on the different aspects of the paper. While both the reviewers recognize the technical integrity of the paper and the clarity of the presentation, the second reviewer's concerns regarding the exclusion of wind retrievals from our measurements and not adapting newer and more sophisticated methods are recognized and responded above. The applications of CMV estimations for guiding the MiniMPL scans and solar irradiance forecasting are also mentioned in the final section. We have revised the Introduction and the Discussion section to clarify these concerns (See annotated file). The title is now modified for more clarity to read "Optimizing cloud motion estimation on the edge with phase correlation and optical flow". The author order has been changed as per their contributions in the current version of the paper.
Revisions to the paper have significantly improved its clarity and quality of contents, and we hope it meets the criteria for publication in AMT.